

# A comprehensive dataset of vegetation states, fluxes of matter and energy, weather, agricultural management, and soil properties from intensively monitored crop sites in Western Germany

Tim G. Reichenau[1], Wolfgang Korres[1], Marius Schmidt[2], Alexander Graf[2], Gerhard Welp[3], Nele Meyer[3,4], Anja Stadler[5], Cosimo Brogi[2], Karl Schneider[1]

[1]Institute of Geography, University of Cologne, Germany
[2]Institute of Bio- and Geosciences 3: Agrosphere (IBG-3), Jülich Research Centre, Jülich, Germany
[3]Institute of Crop Science and Resource Conservation (INRES), Soil Science and Soil Ecology, University of Bonn, Bonn, Germany
[4]Department of Forest Sciences, University of Helsinki, Helsinki, Finland
[5]Institute of Crop Science and Resource Conservation (INRES), Crop Science, University of Bonn, Bonn, Germany

*Correspondence to*: Tim Reichenau (tim.reichenau@uni-koeln.de)

**Abstract.** The development and validation of hydroecological land-surface models to simulate agricultural areas requires extensive data on weather, soil properties, agricultural management, and vegetation states and fluxes. However, this comprehensive data is rarely available since measurement, quality control, documentation and compilation of the different data types is costly in terms of time and money. Here, we present a comprehensive dataset, which was collected at four agricultural sites within the Rur catchment in Western Germany in the frame of the Transregional Collaborative Research Centre 32 "Patterns in Soil-Vegetation-Atmosphere-Systems: Monitoring, Modelling and Data Assimilation" (TR32). Vegetation-related data comprises fresh and dry biomass (green and brown, predominantly per organ), plant height, green and brown leaf area index, phenological development state, nitrogen and carbon content (overall > 17000 entries), and fluxes of carbon, energy, and water (> 180000 half-hourly records) for a variety of agricultural plants. In addition, masses of harvest residues and regrowth of vegetation after harvest or before planting of the main crop are included (> 250 entries). Data on agricultural management includes sowing and harvest dates, and information on cultivation, fertilization and agrochemicals (27 management periods). The dataset also includes gap-filled weather data (> 200000 hourly records) and soil parameters (particle size distributions, carbon and nitrogen contents; > 800 records). This data can also be useful for development and validation of remote sensing products. The dataset (Reichenau et al., 2019) is hosted at the TR32 database (https://www.tr32db.uni-koeln.de/data.php?dataID=1886).

## 1 Introduction

System states and processes at the land surface are of major interest in the context of climate change, hydrological and biogeochemical research. In order to understand the processes in their spatial context and to provide information for larger areas, remote sensing and simulations are heavily applied methods. In this context, it is crucial to understand the fluxes



mediated by the vegetation at the land surface. Dependencies of processes on vegetation states and properties and on environmental conditions are often investigated using models, while their spatial variability is inferred using remote sensing techniques. In this context, well-documented and quality controlled, comprehensive field measurements of vegetation-related variables are essential for research tasks like model development, calibration, parameterization, and validation or as ground

truth for remote sensing products. These variables include biomass per organ differentiated between living (green) and senescent or diseased (brown) material, leaf area index (LAI) and the phenological state of the vegetation. For a simulation, additional information on site conditions such as vegetation composition, soil texture, weather, and, in the case of agro-ecosystem models, agricultural management is required (Kersebaum et al., 2015). However, there is a scarcity of such datasets (Jones et al., 2017). With the publication of the data described in this article, we contribute a new coherent dataset on agro-

ecosystems that includes all of the mentioned variables. The data was collected on conventionally managed fields cultivated by ordinary farmers working at the sites for many years. Thus, it represents conditions and usual practices representative for the intensively used agricultural region to the west of Cologne in Germany. The dataset comprises data from four sites. It consists of almost 1500 records of vegetation parameters and more than 200000 entries of weather data complemented by 15 flux datasets (eddy covariance), management information for 27 management periods, and soil information for all four sites.

Since collecting field-data is very time consuming and expensive there are not many datasets of this size.

The data was collected in the Rur catchment, located at the Belgium-German-Netherlands border within the frame of the Transregional Collaborative Research Centre 32 (TR32, Vereecken et al., 2010, Simmer et al., 2015) "Patterns in Soil-Vegetation-Atmosphere-Systems" funded by the German national science foundation (Deutsche Forschungsgemeinschaft, DFG). TR32 ran from 2007 until 2018. The project's main focus was on the combination of Monitoring, Modelling and Data

Assimilation to assess the role of patterns in Soil-Vegetation-Atmosphere-Systems across scales. The monitoring efforts of TR32 were accompanied by the long-term research program TERENO (Terrestrial Environmental Observatories) of the Helmholtz Association (Bogena, 2016), which made additional instrumentation available for TR32.

Here, we describe the observation sites, the structure of the dataset, and provide information on the observation and measurement methods. Furthermore, we illustrate the quality assurance procedures. With the provision of this dataset, we want

to document our measurement and quality control strategy and provide the scientific community with a comprehensive dataset for further applications.

## 2 Observation sites

All observation sites are located within the Rur catchment located at the Belgium-German-Netherlands border (Fig. 1). The catchment is divided into a fertile loess plain ("Jülicher Börde" and "Zülpicher Börde") in the north and the low mountain

range of the Eifel in the south. The fertile loess plain has a mean elevation of about 100 m above sea level. The land use here is 47 % arable land with the main crops winter wheat, sugar beet, and maize. The warm temperate mid-latitude climate has an annual precipitation of about 700 mm and mean annual air temperature of about 10 °C. The major soils are Haplic Luvisols



and Cumulic Anthrosols near the drainage lines, both with silty loamy textures. Soils close to the river Rur are Gleysols and Fluvisols with silty loamy and loamy sandy textures.

The low mountain range in the southern part of the catchment is characterized by a rolling topography. With a mean elevation of about 690 m above sea level and a mean annual precipitation of about 1400 mm, it is dominated by forest and grassland.

The major soils are Fluvisols, Gleysols (along the Rur and its tributaries), Eutric Cambisols and Stagnic Gleysols with a silty loamy texture.

Location and numbering of the sites and fields are shown in Fig. 1. Terrain properties of each field are given in Table 1. Permissions to take samples from the fields were given by the respective farmers.

## 2.1 Selhausen

The intensively used cropping-site Selhausen is located in the east of the fertile loess plain (50°52'00" N 6°27'01" E). Crops are grown on gentle slopes (0 to 4 %). The altitude ranges from 102 until 110 m a.s.l. According to the IUSS Working Group WRB (2015), main soil reference groups are (gleyic) Cambisol and (gleyic) Luvisol. A westbound dip terrace slope cuts through the site with a NNW-SSE strike separating areas with little gravel in the west (fields SE F01, F02, F10, Fig. 1) from areas with more gravel (fields SE F04, F05, F11 – F14). Fields SE F03, F06 – F09, and F15 show a high content of gravel in

the east but low content in the west.

The climate exhibits an annual precipitation of 698 mm and a mean annual temperature of 9.9 °C (average for 1961–2008, German weather service station Juelich Kernf.-Anlage, Stat-ID 2474, about 5 km north-west).

The Selhausen site was equipped with eddy covariance stations and meteorological sensors since 2007. Because it is the main agricultural observation site of TR32, numerous ancillary data from the site is available and was presented in the literature

(e.g. Busch et al., 2014; Hoffmeister et al., 2016; Korres et al., 2010; Prolingheuer et al., 2014; Schiedung et al., 2017; von Hebel et al., 2018; Bornemann et al., 2011; Ney and Graf, 2018; Schmidt et al., 2012).

## 2.2 Merken

The Merken site (5°50'47" N 6° 24'04" E) is located 4.5 km to the south west of Selhausen. Therefore, soil texture and meteorological conditions are similar. The area is dominated by agricultural fields. The elevation ranges from 107 to 115 m

a.s.l. with slopes of less than 1°. The groundwater at the site is heavily influenced by a nearby open-pit mine. Additional information on the site is presented by Graf et al. (2011).

## 2.3 Merzenhausen

The Merzenhausen site (50°55'47" N 6°17'46" E) is located 13 km to the north west of Selhausen at an altitude of 105 m with a slope of less than 1°. The area is dominated by agricultural fields. Mean annual temperature is 9.7 °C and mean annual



precipitation is 750 mm. The soil at the sampling location is described as an Orthic or Haplic Luvisol (Heitmann-Weber et al., 1994; Schulz, 2004).

## 2.4 Hürtgenwald

The observation site Hürtgenwald (50°43'26" N 6°22'8" E) is located in the northern part of the low mountain range of the Eifel. The altitude ranges from 360 to 375 m a.s.l. with varying slopes. The hilly terrain is dominated by forest, pasture and arable land. The reference soil groups are described as Cambisol or Arenosol (Geological Survey of North Rhine-Westphalia). According to long-term private meteorological measurements (www.huertgenwaldwetter.de), the annual precipitation is 946 mm (2000–2018) and the annual mean temperature is 9.4 °C (1998–2018).

## 3 Conventions and dataset structure

The vegetation data is structured in management periods, which are defined by a combination of the observation site, the field, the crop and the year. A dataset identifier is assigned to each management period as, for example, "SEF05WW15" which describes a management period at the site Selhausen (SE) on field five (F05) where winter wheat (WW) was harvested in the year 2015 (15). A management period can be either the growing period of a crop or the between-cropping period, where the field is fallow. The fallow period can be discontinuous and refers to the periods before planting and after harvesting of a crop.

Data on fluxes and agricultural management are identified by the site, the field, and the year and can thus be assigned to the management periods. Meteorological data is given per site. Soil parameters are available for several points at a site. All measurement locations are identified by their positions and are assigned to fields. Fields are defined by field boundaries with a specific land use and homogeneous agricultural management. In the dataset and throughout this text, sites and land-use types are abbreviated as shown in Table 2, while the field numbering is shown in Fig. 1.

Additional conventions:
- For a crop, the given year is the year when the crop was harvested.
- Throughout the dataset, the symbol NA is used for missing or unknown data.
- Time and date are in UTC.
- Coordinates are given in UTM (Zone 32N, WGS84)

The dataset (Reichenau et al., 2019) is provided as a zip-File containing text files in a separate folder for each site as shown in Fig. 2. Details on the data format are described below. An overview of management periods and available data is presented in Table 3.



# 4 Vegetation data

## 4.1 Data source and methods

The vegetation data contains information on fresh and dry biomass, development state, growth height, canopy density, row spacing, and tissue nitrogen and carbon content. Data on biomass is either given differentiated by organ (brown and green

leaves and stems, respectively, and fruit) or undifferentiated as overall aboveground biomass (named "biomass_undiff"). Furthermore, data on the undifferentiated biomass categories "harvest residues" or "green sprouts" may be included in a record. Harvest residues are understood as the aboveground residues after harvesting which can be material lying on the ground or stubbles left standing. Green sprouts are defined as plants growing between the harvest residues or on an otherwise fallow field. This can be weeds or regrowing crops (especially cereals). In addition, an undifferentiated biomass-category named

"biomass_other" may contain biomass of roots, weeds or the like (specified in the database-column "other_descr").

Vegetation data was collected from 2007 to 2017 on different sites and fields (see Table 3). Biomass and leaf area from at least three points in the field were determined destructively. For row crops, also the number of plants in a certain distance of the row was determined. For cereals, plants were taken from 40 or 50 cm in three different rows. Triticale in Hürtgenwald was not sown in rows. Thus, plants from an area of at least 40 x 40 cm were collected. For crops with large individual plants like maize

or sugar beet, and for rapeseed, the number of plants per square meter was determined from the row spacing and the number of plants per meter. At least three individual plants were collected at each point. In the field, canopy height and row spacing were measured at each sampling location before cutting the plants. The position in the field was determined using a GPS device. In addition, the phenological development state of the crop was assessed using the BBCH scale (Meier et al., 2009).

After being transported to the lab in airtight bags, the fresh weight (FW) of the plant sample was determined. An aliquot of

20 150 g or at least one individual plant was further analyzed. In case of a per-organ analysis, the sample was separated into fruit (understood as the harvested organ like ear, beet, etc.), green or brown stems (shoots), and green or brown leaves. A leaf or stem was classified as brown in case 50 % of its surface were not green. A functional definition of a leaf was applied for cereal leaves where only the leaf blade was considered as leaf, while the leaf sheath was assigned to the stem. Blossoms were defined as fruit. For Maize, the male blossoms on top of the plant were assigned to the stem, and only the female blossoms and the

25 maize cobs that evolve from them were defined as fruit.

The leaf area was determined using either a LI-3000A Area Meter with a LI-3050A Belt Conveyer (LI-COR Biosciences, Lincoln, NE, USA) or a flatbed scanner (Epson GT-15000, Seiko Epson Corp., Suwa, Japan) together with the public domain image analysis software ImageJ (https://imagej.nih.gov/ij/). In a comparison using the same samples, both methods were shown to give equivalent results. Before determining the dry weight (DW), samples were dried in a drying oven at 105 °C for

at least three days. For some samples, aliquots of the dried plant material were homogenized in a mortar and subsequently ground in a ball mill to determine the total content of carbon and nitrogen with an elemental analyzer (CNS Elemental Analyzer Vario EL, Elementar Analysensysteme GmbH, Hanau, Germany). This also includes nine records of C- and N-contents of harvest residues. Upscaling to a square meter of the field was accomplished in a two-step process: from the weighed aliquot





to the sample collected in the field and from the sample to a square meter of the field based on the harvested area or the plant density (for MA, SB, RA). Dry weight and LAI were scaled up in proportion to fresh weight.

Additional information:

- In the years 2015 until 2017, vegetation data was sampled on overflight days of a radar satellite (Sentinel 1, Radarsat 2).

- Per-organ data of crops for fields at a particular site without organ-specific measurements may be estimated from organ-specific biomass measurements for fields of the same crop on this site assuming equal proportions of the total above ground biomass. The validity of this approach depends on similarity of soil and management conditions.

- Prior to 2011, harvest residues and green sprouts were not sampled in the field. Therefore, these entries are always set undefined (NA) in the years 2007 until 2010. LAI is undefined instead of zero, where no LAI was reported in the field protocol.

- During the management periods HWF04HR15 and SEF04HR16, the fields were fallow. Therefore, all biomass is zero. These entries are included in the dataset to document dates, where the field observations showed no biomass on the field. Explicitly distinguishing no biomass from undefined / no data (NA) provides important information for calibration or validation of remote sensing products.

Fig. 3 exemplarily shows dry weights and leaf area index of winter wheat from field F08 at the Selhausen site in 2009 (dataset identifier SEF08WW09). For this management period, three samples per field were collected at each of the 14 dates beginning in December 2008. The last samples were taken on 2009-07-27, one day before harvest on 2009-07-28. The graphs nicely show that the exponential growth phase in April comes along with higher variability between the points in the field in terms of green biomass and LAI. With the beginning of senescence in late May, brown biomass and LAI emerge, showing even higher variability. This is a result of small-scale spatial variability of soil and vegetation properties and terrain under field conditions, which is important information for model evaluation

## 4.2 Quality assurance

The first step of the quality assurance procedure for the vegetation data was a rigorous documentation of the measuring process. In addition to written documentation on any phenomena, which might have affected the measurement (in the field and in the lab), a photographic documentation of the samples in the field and in the lab enables a visual inspection and provides independent evidence in case of any doubts. Transcribing the analog protocols into a spreadsheet-based (MS Excel) digital field protocol provides a first test of data consistency. Possible errors, inconsistencies or incomplete data are reported automatically and the personnel entering the data is prompted to check the entries. Transcribing the data from the analog protocol to a digital data set is done as soon as possible to be able to trace possible errors. Keeping analog field protocols provides a double documentation of the valuable measurements and observations. In a second step, tests on consistency and plausibility were applied which ensure that

- coordinates are in UTM projection, and timestamps are in UTC,




- naming of crops, sites, and points follows conventions,
- values are in plausible ranges,
- missing values are set unknown (NA),
- the right upscaling method is set for a crop throughout a management period,
- there are no duplicate coordinates for points in a field at the same date.

A third step comprises statistical tests, which result in a quality flag for each value in the dataset (see below). These tests were applied using an R-Script (R Core Team, 2017) which reads from the digital field protocols, assigns the quality flags, and finally writes the files provided in the dataset.

**Quality flags**. The quality flags can take the values 1 to 5:

1. High quality (all tests could be applied and no problems were identified; no problems were identified in the field)

2. Good quality (a test could not be applied; information is missing to ensure high quality)

3. Unusual water content (a specific flag concerning the measured water content of the sample which may hint at problems with biomass measurements)

4. Suspicious (a test or a documented issue in the protocol showed possible problems)

5. Low quality (a value is known to have problems but is of interest as an evidence of the real conditions, e.g. root biomass)

The flags were set based on the criteria explained below. After evaluation of all tests, the flag with the highest value was assigned. Obvious erroneous data was removed from the dataset. There are no flags for the carbon and nitrogen contents of the plant tissue.

**General flagging**. Weight measurements below 1 gram were generally flagged as good quality (2) instead of high quality (1),
as it is quite likely to lose material from samples, which will have a larger relative effect than for high biomass. All harvest residues are generally flagged suspicious (4). This is due to the fact that precise collection of only the aboveground material is rather difficult and error prone. It is even more difficult to extract the below ground biomass. Therefore, root biomass (given as "biomass_other") is generally flagged as low quality (5).

**Loss of material**. In most cases, a sample from the field had to be differentiated into fractions (organs, harvest residues, green
sprouts). For larger samples, only a part (aliquot) was analyzed in the lab (see section 4.1). For organ-specific analysis, this aliquot is the sum of all organs. In case of undifferentiated biomass, the aliquot is the sum of the biomass categories biomass_undiff, harvest_residues, green_sprouts and biomass_other. During the process of sample partitioning some material might get lost causing a difference between the aliquot and the sum of its components (median 1 %). Differences up to 5 % were accepted independent of their sign. Larger differences result in higher values of the quality flag (Table 4). Higher flags
are set in case the sum of its components exceeded the aliquot, because this cannot be explained by losing material.

**Reconstruction of missing values.** If an aliquot was available but the FW of one of its components was missing, this FW was recalculated from the difference of the aliquot and the sum of the available FWs. Due to the missing value the loss of material during sample partitioning cannot be determined. Instead, it is contained in the recalculated value, which is therefore flagged





as suspicious (4). In this case, the test against the aliquot is not applicable. Thus, the other FWs were flagged as good quality (2).

**Comparison of fresh and dry weight.** The comparison of FW and dry weight (DW) can reveal errors in the biomass data. In a first step, it was tested whether DW exceeds FW (Figure 4). For brown leaves and stems, FW and DW were compared

directly, while for the other biomass categories, for this test the FW was reduced by 5 % assuming that percentage of minimal water content. In case DW exceeded the resulting FW, it was checked whether the sum of fresh weights was less than 95 % of the aliquot, which hints at a possible error in the FWs (see above). In that case, the error cannot be attributed to neither FW nor DW and both were removed from the dataset. If the sum of fresh weights was more than 95 % of the aliquot, the error was attributed to the DW, which consequently was removed from the dataset and the corresponding FW was flagged good quality

(2).

In a second step it was checked whether the relative water contents (FW-DW)/FW of green stems, green leaves and fruit are within the range of usual values. This can hint at problems with the DW and FW, which were not identified based on either of the weight values alone. At first, it was assumed, that living plant tissue has at least a water content of 50 % and DW and FW of green stems or leaves were flagged suspicious (3) if the relative water content was below 50 %. In addition, a "usual course"

of the relative water content (Fig. 5) was defined for fruit, green leaves and green stems for winter wheat, winter barley, rapeseed, maize, and sugar beet, respectively. In order to define a lower and upper boundary of the usual water content, the following steps were executed:

1. Use all water content data for a respective crop and organ.
2. Exclude outliers by removing all values outside of the 10 % and 90 % percentiles in a running 21-days window.

3. In each time window, determine the corridor of two standard deviations above and below the mean.

Owing to the low number of data for some crops and organs, and to its scattering, the upper and lower boundaries of the corridor show a lot of scatter. Since there is tendency towards lower water content with progressing phenological development, the limits of the usual course were defined as follows:

4. Lower limit: For each day in the direction of time only include the lower boundary of the corridor, if it is lower than the

value at the previous day. Otherwise, keep the value of the previous day as the lower limit of the usual course.

5. Upper limit: For each day in reverse direction of time only include the upper boundary of the corridor, if it is higher than the value at the following day. Otherwise, keep the value of the following day as the upper limit of the usual course.

For water contents outside of the upper or lower limits, FWs and DWs were assigned the "unusual water content" flag (3). However, this data might also result from particularly dry or wet conditions at a point in a field in a certain year.

**Reported issues.** All issues observed in the field or in the lab which may have had an influence on the results, were translated into flags. For samples reported as dirty, FW and DW were flagged suspicious (4). For humid or wet samples, samples, which might not have been completely dried, and samples which were not analyzed on the same day, only the FW was flagged suspicious (4) since DW is not affected. In case the number of plants per meter was required for upscaling (MA, SB, RA) but missing, this value was derived from other points or dates in the same management period and field. Since this propagates





linearly to LAI and to all biomasses per square meter and, since the germination rate is variable in space, all FWs and LAI were flagged suspicious (4).

**Propagation of quality flags.** FW and DW are connected by the upscaling process from the aliquot to the sample (see section 4.1), because the upscaling factor derived from FW is also applied to DW. Therefore, flags were propagated from FW to DW

and in case of leaves also to LAI.

**Coordinates.** To ensure the validity of the location coordinates it was ensured that reported coordinates of a given measurement are within the given field and that no duplicate coordinates are assigned to different measurements at the same date. If it was not possible to correct implausible coordinates, they were removed. In 2008, measurement locations within each field were predefined and marked with flags. Consequentially, coordinates were not recorded explicitly. Since destructive

sampling employed in this study prevents repetitive sampling of the exact same location, the prescribed coordinates represent the sampling location less accurate than those recorded directly at the sampling points. Thus, coordinates for 2008 were flagged good quality (2) instead of high quality (1).

### 4.3 Uncertainty

Uncertainty of biomass data is difficult to estimate. Sources of error exist in all steps of sampling and analysis, including

15 harvest of the samples in the field (incomplete harvest), loss of material and water during handling of the sample, and the unsystematic error of the scales. The error of incomplete harvest cannot be quantified based on the existing data. However, the relative error can be assumed to be rather small for high biomass. The error of handling the sample in the lab (separation of the sample) can be assessed by comparing the weight of the aliquot that was separated by organ with the sum of the organ-weights. Of 1176 organ-specific records in the dataset, 229 have a valid aliquot. The other records either show missing values,

or do only have a single organ, or were weighed in total without taking an aliquot. 164 records show a loss of material during separation, while 20 show an increase. The mean loss is 2.6 % of the aliquot (max. 15 %). The mean increase is 2.9 % (max. 17 %). The average error for the (un-)packing steps associated with transport and drying cannot be quantified based on the available data. However, since activities are similar, it can be assumed to be of a similar relative magnitude. The maximum error of the scales used in the lab was 0.1 g. Since leaf area can be measured quite precisely, the relative uncertainty of LAI

depends primarily on the accuracy of the leaf weight used for upscaling. Since these are connected linearly in the upscaling process, it equals the relative uncertainty of biomass. A further source of error is the upscaling from the sample taken in the field to a square meter. For row crops (see section 4.1), the error of the measured row spacing or plant density within the rows propagates linearly into the upscaled result. In order to reduce this error, the median of all row distances measured on a field in a management period was used for upscaling. As the sowing machine settings do not change within a field, the resulting

error is considered small. In the field, plant height was measured with a folding rule. The reading accuracy is assumed to be 1 cm which is less than the natural variability of plant height.

The uncertainty of carbon and nitrogen contents of the plant tissues was determined by analyzing differences of 1034 duplicate measurements (two aliquots of the same sample). For carbon content, the mean difference of the samples was 0.6 %. For nitrogen content, the mean difference was 1.1 %.The largest differences occurred for root tissues.

Concerning the uncertainty of phenological states in the BBCH system, principal growth stages (first digit) can be assumed to
be correct, while secondary growth stages (second digit) may have an error. Since this depends on the observer, it cannot be generally quantified.

### 4.4 Data format

Vegetation data is supplied per site in a UTF-8 coded csv-file named "vegetation_" followed by the two-letter site abbreviation (Table 2). Column separator is the semicolon (;). A description of columns and units is presented in Table A1. The no-data
symbol is NA. The files have two header lines, of which the first contains the variable names while the second contains the units.

Phenological development (column bbch) may be given as a single number or as a range, if the development state could not be exactly identified in the field. Before sowing and after harvest, the land-use is set to harvest residues (HR) independent of the presence of residues on the surface of the field.

**5 Fluxes of carbon, water and energy**

### 5.1 Data source and methods

The dataset contains 15 time series of flux measurements. Net fluxes of carbon (NEE), water (LE), and energy (H) at the surface were measured at the sites Selhausen, Merzenhausen and Merken using state of the art eddy covariance systems. Wind components and sonic temperature were measured with a three dimensional sonic anemometer (CSAT3, Campbell Scientific,
Inc., Logan, UT, USA). Measurements of water vapor ($H_2O$) and carbon dioxide ($CO_2$) density were carried out using an open-path infrared gas analyzer (IRGA, model LI7500, LI-COR Inc., Lincoln, NE, USA). In Merken, each EC-tower was equipped with two sets of sensors at different heights (Table 5). The lower measurement height is usually more representative of the respective land use type. However, the upper level has provided an even better energy balance closure, than the already good one of the lower level.
Measurements were taken with a sampling rate of 20 Hz and were aggregated to intervals of 30 minutes. Processing of raw measurements was accomplished as shown in Fig. 1 of Mauder et al. (2013) using the processing software shown in Table 5. The number of decimal places in the datafiles were kept as they were in the output of the processing software.

### 5.2 Quality assurance

Quality control was accomplished according to the "TERENO" scheme for quality and uncertainty assessment presented by
Mauder et al. (2013). Deviating from this description, before 2011 the software TK2 (Mauder and Foken, 2011) was applied



following the process described in section 2.3 of Schmidt et al. (2012). The software ECpack 2.5.20 (Van Dijk et al., 2004) was applied for the data from Merken (Table 5). The software TK uses flagging to indicate the quality of data. Flag values and their meanings are shown in Table 6.

Since flux data from Merken 2009 (MK09) was processed with the ECpack software, the concept of quality assurance differs from the other sites. ECpack provides tolerance values which can be used to rate the quality of data (Table 7). Values outside the lower and upper boundaries given in Table 7 are considered invalid. In addition, data can be filtered using the tolerance values. A tolerance is assigned to the lower and upper boundary of each variable, respectively. To evaluate the quality of the data in the valid ranges, tolerances have to be linearly interpolated between the boundaries. The most obvious tolerance violations have already been eliminated by a post-processing scheme. Tolerance limits were set sufficiently wide to retain most of the values, which still might be useful, in the dataset. For some variables, considering a value to be invalid causes the whole record to be invalid. These variables are assigned to group A in Table 7. If any value of group B is considered invalid, only the values of group B are invalid.

### 5.3 Uncertainty

Uncertainty information for fluxes per data-point is available for sensible heat flux, latent heat flux, NEE, and friction velocity. The kind of uncertainty information differs between the different software tools used for data processing (Table 5). For TK3, relative random errors and relative noise errors for friction velocity, sensible and latent heat flux, and for net ecosystem exchange are given in the respective columns (see Table A3) in the datafiles. For datasets processed with TK2 this information is not available. A rough estimate of the general uncertainty for these measurements may be obtained from statistics of the errors included given in the TK3-processed data. For other variables included in the TK output, the uncertainty is quantified from the instrument errors given by the respective manufacturers (Table 11). The uncertainties of $CO_2$ and water contents of the air (variables a and CO2) strongly depend on calibration. Detailed information can be obtained from the manual of the infrared gas analyzer (LiCOR LI7500, LI-COR Inc., Lincoln, NE, USA). However, the accuracy of the absolute measurements is of minor importance for the eddy-covariance method since it depends on relative changes. The other software tool, ECpack, calculates 95 % confidence intervals per data-point for fluxes and several other variables. These so-called tolerances are given in the respective columns (see Table A2) in the datafiles. Additional information on uncertainties of eddy-covariance measurements is presented by Mauder et al. (2013).

### 5.4 Data format

Flux data is provided in a UTF-8 coded csv-file per field and year. The filename consists of "fluxes" followed by the two-letter site abbreviation (Table 2), the field ID (Fig. 1), "EC", a station identifier, and the year. The elements of the filename are separated by underscores (e.g. fluxes_SEF01_EC_001_2016.csv). Column separator is the semicolon (;). A description of columns and units is presented in Table A2 and Table A3 for the TK and ECpack software, respectively. The no-data symbol is NA. The files have two header lines, of which the first contains the variable names while the second contains the units.





## 6 Soil properties

### 6.1 Data source and methods

Soil property data includes particle size distribution of the fine soil (<2 mm), proportion of coarse material (gravel, >2 mm), bulk density, and soil carbon and nitrogen contents. The availability of data differs from site to site (Table 8).

All particle sizes were analyzed following DIN ISO 11277. Therefore, they follow the definition of particle size classes of DIN 14688. Particles larger than 2 mm are considered gravel. To recalculate particle sizes to the USDA system, which is assumed for many pedotransfer functions, refer to e.g. Nemes et al. (1999). All data on particle sizes and soil carbon or nitrogen-contents refers to fine soil after the removal of gravel. Therefore, percentages of sand, silt, and clay refer to fine soil, while the percentage of gravel refers to total soil mass. Bulk density was determined gravimetrically. Total C concentrations

in soil samples were determined by elemental analysis. Based on previous analyses it can be assumed that all samples were free of carbonates. Hence, total C concentrations are in accordance with those of SOC.

**Selhausen**. Soil data for Selhausen originates from different sources. Particle sizes for three depths in field SE F08 were analyzed at the Laboratory for Physical Geography, University of Cologne. For the ploughing horizon of field SE F00, particle sizes were analyzed at the Institute of Crop Science and Resource Conservation, Bonn University (Bornemann et al., 2011).

This data has a high spatial resolution that enables analysis of small-scale heterogeneity. A third dataset consists of horizon-specific particle size data from 100 randomly chosen points from a 1 km² area that includes most fields with vegetation data. The samples were analyzed at the Soil Physical Laboratory of IBG-3, Jülich Research Centre, using a Sedimat 4–12 apparatus (UGT, Umwelt Geräte Technik GmbH, Münchenberg, Germany). From this data and extensive EMI measurements, Brogi et al. (2019) generated a map of soil units, which groups the abovementioned 100 sampling locations into 18 geophysics-based

soil units composed of two to twelve sampling locations. These soil units are also provided with quantitative description (layering, texture, total carbon and nitrogen content) of the soil profile. In the files containing information on the soil and vegetation samples, a column (soil_unit) establishes the link to the respective soil unit.For several fields, total carbon and nitrogen contents for three depths were determined from composite samples at the Laboratory for Physical Geography, University of Cologne, using a CNS Elemental Analyzer (Vario EL, Elementar Analysensysteme GmbH, Hanau, Germany).

If data was available for several dates, a date after harvest but before the next fertilizer application was preferred if possible. This is noted in the comments column of the data-table. Soil carbon and nitrogen data are assigned to a field instead of a specific location, because a composite sample containing equal fractions of material from several points was analyzed. From the 100 sampling points of the 1 km² area, carbon and nitrogen content for two horizons, (Ap and Bw) were determined for composite samples from all sampling points within a soil unit, respectively. Therefore, this data is given per soil units. To

determine nitrogen and carbon content, a standard combustion method was used at the Geography Institute of the Ruhr University Bochum using a CNS Elemental Analyzer (Vario Max, Elementar Analysensysteme GmbH, Hanau, Germany). All samples were collected between 6th and 15th February 2017. It has to be noted that samples were collected regardless of the agricultural management.

Due to temporal and spatial variability, this data has to be understood as snapshots and cannot be transferred to other points in space or time.

**Merken**. Particle size data for the Merken site is only available from a composite sample based on samples from all fields. This data is assumed to be valid for all fields due to small spatial heterogeneity of the soil at the site. The analysis was carried out at the Soil Physical Laboratory of IBG-3, Jülich Research Centre. Field-specific carbon and nitrogen contents for three soil depths were measured from composite samples as described for Selhausen.

**Merzenhausen**. For the field in Merzenhausen, soil texture, bulk density, and contents of carbon and nitrogen were determined for the Ap horizon at a single point in the field following the methodology described by Bornemann et al. (2011). Bulk density was quantified from three independent 100 cm³ samples. Since no data was collected for other soil horizons in the frame of the TR32 project, we include data published by Pütz (1993) for the sake of completeness.

**Hürtgenwald**. For Hürtgenwald, particle sizes were analyzed at the Institute of Crop Science and Resource Conservation, Bonn University, while bulk densities were determined at the Laboratory for Physical Geography, University of Cologne.

## 6.2 Quality assurance

For the determination of particle sizes, bulk density, and carbon and nitrogen contents, at least two samples from each point were analyzed in parallel. This was not the case for the 1 km² data from Selhausen where single analyses were carried out. In this case, the weight of the sample was taken before and after the texture analysis. The analysis was repeated if the final weight was lower than 95% of the initial weight. If at the second iteration the value was again lower than 95%, the analysis was repeated for a third time.

## 6.3 Uncertainty

To quantify the uncertainty of particle size fractions, data of a repeatedly analyzed sample was evaluated at Bonn University. The results show coefficients of variation (CV) of 2.0 % for sand, 2.4 % for silt, 2.5 % for clay, and 3.5 % for gravel. Since such repeated estimates were not performed at the University of Cologne, it is assumed, that the uncertainty of their measurements is of the same magnitude. At Jülich Research Centre, particle sizes were automatically analyzed with a Sedimat (see above), which has uncertainties in the calculation of the particle size fractions that are comparable to those obtained in above mentioned analysis performed in Bonn. For bulk density, a CV of 10 % was determined from the analysis of multiple adjacent samples from the same horizon (Bonn University). For the soil unit data from Selhausen, uncertainties for particle size fractions and layer depth are given in the respective columns (Table A6) in the datafiles. The CNS Elemental analyzers used to determine soil carbon and nitrogen contents show uncertainties of ±0.01 % for carbon and ±0.002 % for nitrogen.

## 6.4 Data format

Soil data is provided in a UTF-8 coded csv-file per site named "soil_" followed by the two-letter site abbreviation (Table 2). Column separator is the semicolon (;). A description of columns and units is presented in Table A5. The no-data symbol is





NA.Soil unit data for Selhausen is provided in a UTF-8 coded csv-file named "soil_units_SE.csv". Column separator is the semicolon (;). A description of columns and units is presented in Table A6. The no-data symbol is NA. The files have two header lines, of which the first contains the variable names while the second contains the units.

## 7 Meteorological data

### 7.1 Data source and methods

The dataset was assembled with the aim to provide the data usually required to run a hydroecological crop growth model. Therefore, the dataset includes gap-filled hourly meteorological data of air pressure (AirPres, Pa), air temperature (AirTemp, K), relative air humidity (AirHum, %), wind speed (Wind, m s$^{-1}$), precipitation (Precip, mm h$^{-1}$), cloudiness, (Cloudiness, 1/8), and global radiation (Globrad, W m$^{-2}$). The meteorological data starts about one year earlier than the vegetation data, to provide data for model spin-up concerning water pools in the vadose zone.

The availability of meteorological field data varies between the sites as well as in time. The temporal availability of data increased significantly with time due to the setup of the meteorological stations of the TERENO Eifel/Lower Rhine Valley observatory in 2011. In the earlier years, only a few meteorological stations were run near the sites. Table 9 shows a list of all meteorological stations used in this study. Methods to fill gaps in the time series vary between years and stations. The gap-filling methods are explained in the following sections. The data source for each year and site is presented in Table 10. In most cases, information on the measurement devices and raw data with gaps can be obtained from the data sources shown in the table.

Meteorological data for the site **Hürtgenwald** is provided for the years 2014 through 2016. Meteorological measurements started on 21 April 2015 (station 20 in Table 9). In 2016, additional stations were set up nearby (station 21 in Table 9). Data for earlier dates was generated using the regression gap-filling method (see section 7.1.1) for all variables but AirPres, where gaps were filled using the barometric formula (Eq. 1). The first year for HW consists of reconstructed data only.

Data for the site **Merzenhausen** is provided for the years 2009 through 2017. Local measurements are available for the whole period (stations 1 and 15 in Table 9). For the years 2009 and 2011, gaps were filled using the "regression" method. From 2011 on, the EOF method was used (see section 7.1.1).

Data for **Selhausen** is provided for the years 2007 through 2017. Local measurements are available for the whole period, starting on 27 May 2007 (stations 10, 11, and 19 in Table 9). For the years 2007 until 2010, the regression method was used to fill gaps. From 2011 on, the EOF method was applied.

For the site **Merken**, no meteorological data is available. Since the distance to Selhausen is only about 4 km and the difference in elevation is about 10 m, it can be assumed, that the weather was very similar to that in Selhausen. Therefore, it is suggested to use Selhausen meteorological data when working with Merken vegetation or flux data.

Since cloudiness is not available for any of the sites, but required in some ecohydrological models (e.g. the DANUBIA simulation system, for an application see e.g. Korres et al., 2013), data on cloudiness from the German National Weather



Services' station Aachen (distances to HW, ME, SE 37, 37, and 42 km, respectively) was used. Since there is no reliable method to adjust cloudiness data to remote stations, the data was used without modifications.

Information on the conditions at the locations of the meteorological stations, especially in the past, are not fully available. Therefore, precipitation data is given as measured at the stations. Since the data was not corrected for shielding effects, precipitation can be assumed to be slightly underestimated.

Fig. 6 shows an excerpt of the meteorological data for the Selhausen site for the period May to July 2011. The graphs show a period where there are no breaks or shifts in the continuous curves, which is the usual case in the weather timeseries (for a discussion of inhomogeneities, compare section 7.1.3). In the middle of June, the example data shows a noticeable period of two days with low radiation and temperatures together with rather high wind speed and high cloud cover. All variables show a reduced diurnal cycle, which confirms the consistency of the timeseries' of the separate variables, which is an important prerequisite for a good reproduction of real processes in a simulation. On short timescales

### 7.1.1 Gap filling

In the course of the TR32 project, an increasing number of meteorological stations was set up in the Rur catchment. Therefore, different methods were chosen for different periods to fill gaps in the meteorological data.

**Insertion method.** For this simple approach (method 0 in Table 10), data of a nearby station was simply inserted into gaps of the reference station's timeseries.

**Regression method 1.** This method (method 1 in Table 10) was applied to fill gaps in the years 2007 through 2010 in Selhausen and for the whole period in Hürtgenwald. A simple linear regression was set up between the available data of the station with gaps and a nearby station for each variable, respectively. The slope of the regression was then applied to the data of the nearby station to fill the gap. In case of a data gap at the nearby station, data from a further station was used. In the seldom cases where no data was available at any station, the gap was filled based on linear interpolation. No gaps longer than four hours had to be filled this way.

**Regression method 2.** For variant 2 of the regression method (method 2 in Table 10), which was applied for the year 2010 in Selhausen and for the years 2009 and 2010 in Merzenhausen, the data of a reference station was correlated with data of the closest remote station using a reduced major axis regression (Webster, 1997). If the coefficient of determination was higher than 0.9 the data of the remote station was inserted into the data gap without further processing (same as insertion method). In case $R^2$ was lower than 0.9, the slope of the regression (for AirTemp also the offset) was applied before inserting data into the data gap. For AirHum, the method was applied to dewpoint temperatures, which were converted back to relative humidity after gap-filling.

**EOF method.** This method (method 3 in Table 10) was applied for the sites Merzenhausen and Selhausen. It utilizes empirical orthogonal functions (EOF) to describe the relation between variables at several meteorological stations. The approach was originally introduced by Beckers and Rixen (2003) and adapted for station time series by Graf (2017); further information on EOF computation on similar data can be found in Graf et al. (2012). Since the approach does not depend on the regular spatial





arrangement of the pixels, it can easily be transferred to a network of stations. In contrast to the original approach, this method works on the z-transform of each time series (normalization by dividing the deviations from the mean by the standard deviation), which ensures that stations where the variable has a low amplitude receive the same importance as a predictor as others with a larger amplitude. The following steps were accomplished for each variable separately. Shortwave incoming

(global) and photosynthetically active radiation, however, were treated jointly due to their close linear relation:

0. Prior to gap-filling, remove all values rated „bad", or „suspicious".

1. Delete an additional 10 % (randomly selected) of the available data per station and set them aside for cross-validation purposes

2. z-transform the data for each station and variable, respectively

3. Replace all missing values by zeroes

4. Compute the EOFs and reconstruct the time series of each station and variable using only the first EOF ("truncated reconstruction")

5. Fill all gaps with the reconstruction and repeat step 4 with the filled time series. Repeat the procedure until no data point is changed from one iteration to the next by more than 1 %, or if the change between iterations starts to increase again in at

15 least one data point, or if a maximum of 1000 iterations is reached.

6. Use the dataset with the new preliminary fillers to initialize at step 4 again, but this time using the first two EOFs. Continue as in step 5. After this has converged too, use the first 3 EOFs and so on, until 10 EOFs are used.

7. Re-transform results to absolute values (reverse step 2).

8. Use the cross-validation dataset set aside in step 1 to determine the number of EOFs at which the prediction is optimal

(minimum RMSE between validation data and prediction). Repeat the whole procedure up to this number of EOFs starting with step 2 (i.e. without removing cross-validation data).

An advantage of this approach is, that the EOF method exploits the same underlying statistics as multiple linear regression would, but does not need to be re-evaluated each time a predictor variable becomes unavailable. The method was applied to 10-minute resolution data from stations 1 to 18 (Table 9). Results were aggregated to hourly resolution.

**Gap-filling of cloudiness data**

Gaps in cloudiness data were filled using the "na.approx" method in the R-package "zoo" (Zeileis and Grothendieck, 2005).

**7.1.2 Adjustment of atmospheric pressure**

For the sites and years where the EOF-Method was not applied, air pressure (AirPres, in hPa) data was transformed between stations by using the barometric formula:

$$\text{AirPres} = \text{AirPres}_r \left(1 - \left(0.0065 \frac{\Delta h}{\text{AirTemp}}\right)\right)^{5.255} \qquad (1)$$

Where $\Delta h$ is the elevation difference between Stations (m), AirTemp is the air temperature (K), and AirPress is the atmospheric pressure at the remote station (hPa).



### 7.1.3 Inhomogeneities

A closer look at the time series of meteorological data reveals differences in general characteristics between different years. This is mainly due to different instruments or different calibration of instruments. By these means, synthetic breaks in the time series are generated that can disturb the analysis of real phenomena. This is particularly a problem when using the data with

models, which deterministically transform weather data into plant growth and into exchange fluxes of matter and energy. Several breaks can easily be identified from graphical visualizations of the data. Fig. 7 a shows a shift in air pressure measured in Selhausen from 2009 to 2010 using different instruments. A similar effect can be observed in the Merzenhausen data. Fig. 7 b illustrates different maxima of relative humidity in 2015 in Hürtgenwald, which are due to differences in instrument calibration. This effect can also be found in the data for Merzenhausen and Selhausen. Other obvious breaks refer to lower

extrema of air temperature (SE and ME), maxima of global radiation (ME), maxima of wind speed (SE), and changing temporal variability of wind speed (HW). Often, these breaks coincide with a change in the main source station (Table 10). Other less noticeable breaks may be included in the time series.

The removal of such breaks in the time series is known as homogenization in the literature. Several methods have been developed to detect the breaks and correct for inconsistencies. However, most of these methods were designed for monthly or

annual data (Venema et al., 2012), and are not applicable to subdaily data (Aguilar et al., 2003, Auer et al., 2005, Wijngaard et al., 2003). Since methods for data on higher temporal resolutions would involve dealing with non-linear atmospheric processes (Della-Marta and Wanner, 2006), the world meteorological organization does not yet make any recommendations on how to homogenize this data. Nevertheless, the following literature might help finding an appropriate homogenization method for the intended application of the data. Temperature: Vincent et al. (2002), Brandsma and Können (2006), Della-

Marta and Wanner (2006), Kuglitsch et al. (2009), Mestre et al. (2011), Trewin (2013); Precipitation: Beaulieu et al. (2008), Beaulieu et al. (2009); Both: Domonkos and Coll (2017).

### 7.2 Quality assurance

Time series of meteorological data were checked for plausibility of the recorded data. Values outside of a plausible range were removed from the dataset. Periods of repeated identical (but plausible) values were removed. To ensure good quality of gap-

filling, the gap-filling methods were applied to periods with good quality measurements.

### 7.3 Uncertainty

Measurement uncertainties of weather variables are given as instrument errors in Table 11. It has to be mentioned that especially for precipitation, the instrument error is much smaller than systematic errors. For a discussion of such errors, compare Dengel et al. (2018).

Additional uncertainty occurs when gaps in timeseries are filled based on data from other stations. Because different methods and data sources were used, uncertainty was determined separately for the different sites and years. For the years 2007 to 2010,



uncertainty was estimated by deriving a fill-value from remote stations for each available value at the reference station using the respective method shown in Table 10. Bias and root mean square error (RMSE) were calculated from the differences (Table 12). These results for the Selhausen site are assumed to be transferable to the other sites.

For the EOF method, which was applied for the years from 2011 to 2017, an extra run on a dataset copy with artificial gaps was used to determine worst-case uncertainty estimates. These artificial gaps were inserted for the Merzenhausen site for the 2.5 consecutive days with the highest mean for the respective variable (relative humidity: lowest mean) for all sensors at the site. The artificial gaps were then filled and the differences to the measured data were evaluated in terms of bias and RMSE (Table 12). By selecting an extreme situation for gap-filling, uncertainties for the EOF-method are a worst-case estimate. Inserting arbitrary gaps would probably give lower uncertainty values. In addition to this, when comparing uncertainty estimates between different periods, it has to be taken into account that the analysis for the EOF method was applied to raw 10-min data while the evaluation for the years 2007 to 2010 is based on hourly data, which generally results in slightly lower RMSE values. Again, we assume that results can be transferred to the Selhausen site.

For precipitation in the period 2007 to 2010 and for global radiation in 2010, uncertainty estimates cannot be given since the raw data is no longer available. Data from the German weather service (DWD) was used for global radiation in 2007 to 2009 and for air pressure in 2010. Since this data was without gaps, there is no gap-filling uncertainty for these variables.

**7.4 Data format**

Weather data is provided in a UTF-8 coded csv-file per site named "meteo_" followed by the two-letter site abbreviation (Table 2) and the span of years available. Column separator is the semicolon (;). A description of columns and units is presented in Table A7. The no-data symbol is NA. The files have two header lines, of which the first contains the variable names while the second contains the units.

**8 Crop management data**

**8.1 Data source and methods**

Crop management data was inquired from the farmer of the respective fields by means of a questionnaire. However, the information given by the farmers is often incomplete. The following information was inquired:

- Sowing date
- Sowing density, row spacing, seed spacing in a row, seed weight, sowing depth, cultivar
- Fertilization date, amount, and product
- Cultivation date and type
- Growth regulator application date, amount, and product
- Fungicide/Insecticide/Herbicide application date, amount and product
- Harvesting date



- Dry weight of yield after harvest
- Information on residues left on the field

All fertilization data was recalculated to kilograms nitrogen per hectare. Since for some products, nitrogen content is not explicitly stated, the following assumptions were made: It was assumed that KAS (calcium ammonium nitrate) contains 27 mass-% nitrogen. Furthermore, it was assumed that Sulfan contains 24 % nitrogen. AHL (urea ammonium nitrate solution, UAN) was assumed to have a density of 1.3 kg l$^{-1}$. All fields were managed conventionally.

## 8.2 Quality assurance

Some of the fields were equipped with automatic camera systems, which took hourly photos. Management information gathered from the farmers were checked against these photos.

## 8.3 Uncertainty

Accuracy of management data is based on the reliability of the information provided by the farmers. Since there is no way to check information on fertilizer or agrochemical types and amounts, an uncertainty cannot be assigned.

## 8.4 Data format

Management data is provided in a UTF-8 coded csv-file per management period. The filename starts with "management_" followed by the ID of the management period (e.g. management_SEF08WW09.txt). The file can contain data on management activities in the fallow period before or after harvest. If no management information is available, the file contains a comment only. There are no management files for management periods denominated "harvest residues" (HR).

Each record is structured in the same way: date; keyword; additional information. The elements of the record are separated by a semicolon (;). The record starts with the date in YYYY-MM-DD format, where day may be replaced by "xx" if the exact date is unknown. In the second position, the record contains a keyword that defines the management activity. Keywords refer to basic crop related activities ("Sowing", "Harvesting", "Fertilizer", "Cutting"), soil management ("Plow", "Rotary harrow", "Harrow", "Roller", "Cultivator", "Tyre Packer"), and application of agrochemicals ("Herbicide", "Growth control", "Fungicide", "Insecticide", "Co Formulant"). After the keyword, one or more pieces of additional information may follow in a semicolon-separated list:

- Fertilizer: amount of fertilizer in kilograms nitrogen per hectare; information on the product and its contents (may also be a semicolon-separated list)
- Application of agrochemicals: amount of agrochemical per area; information on the product and its contents (may also be a semicolon-separated list)
- Sowing: sowing density; row spacing; seed spacing; weight of seeds; sowing depth; cultivar

Unknown information is indicated by the no-data symbol NA. Units are given with the data. Comments start with "#". Comments can contain additional information on yield, management of harvest residues, additional contents of agrochemical, etc.

**Data availability**

The dataset (Reichenau et al., 2019) can be downloaded from the TR32 database (https://www.tr32db.uni-koeln.de/data.php?dataID=1886). The dataset is provided as a zip-compressed container. All files are plain text files organized in a folder per site as show in Fig. 2 and as explained in section 3. Technical details on file formats and data structure within files is presented for the different kinds of data in sections 4.4, 5.4, 6.5, 7.4, and 8.4.

**Appendix:**

The tables in the appendix describe the datafiles in terms of their column order, variables, units, and datatypes.

**Author contributions**

TR designed and compiled the dataset, and did the quality control and processing of the vegetation data. AS and AG did the gap filling of meteorological data and developed the methods. MS processed the eddy covariance data and collected the management data. WK and KS were responsible for the collection of the vegetation data and some soil data. GW, NM, and
CB took and analyzed soil samples. The manuscript was prepared by TR with contributions of all co-authors.

**Competing interests**

The authors declare that they have no conflict of interest.

**Acknowledgements**

This study was supported by the Deutsche Forschungsgemeinschaft through the Transregional Collaborative Research Center
32 – Patterns in Soil-Vegetation-Atmosphere Systems: Monitoring, Modelling and Data Assimilation. In addition, support was received through the "Terrestrial ENvironmental Observatories" (TERENO). We thank the numerous student helpers for their help with the field campaigns. Special thanks go to the farmers who granted access to their fields and to our student helpers Michael Holthausen (gap filling of meteorological data), Tobias Bothe (gap filling and soil sampling), and Nils Eingrüber (consistency check of soil data). We thank Marius Schmidt and Florian Steininger for collecting the management data from



the farmers, Victor Venema for discussions and literature on the homogenization of meteorological, Ulrike Schwedler for cartography, and Constanze Curdt for advice concerning all aspects of data management.

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



**Figure 1: Left: Locations of the observation sites in the Rur catchment in Germany. Right: Locations of the fields at the observation sites with two-digit field IDs. At the Selhausen site (3), field 12 is a part of field 11. On the aerial photo of the Merken site (2), a part of field 01 is on the area of an open pit mine. At the time of field measurements, the mine was about 2.5 km away from the field. Map-data: GADM (gadm.org/license.html), OpenStreetMap (© OpenStreetMap contributors 2019. Distributed under a Creative Commons BY-SA License. ) (Open Database Licence (ODbL) 1.0). Aerial photography: Land NRW (2019) Datenlizenz Deutschland - Namensnennung - Version 2.0 (www.govdata.de/dl-de/by-2-0).**



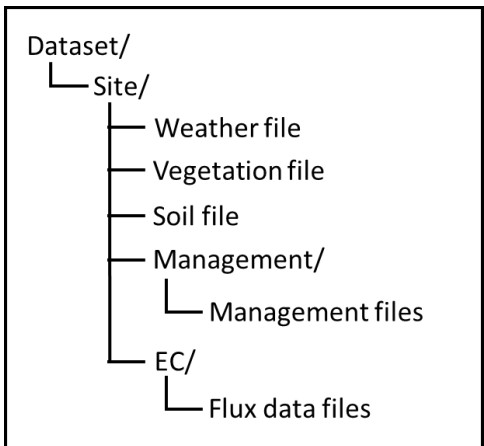

**Figure 2: Folder structure of the dataset. A slash ("/") denotes a directory.**

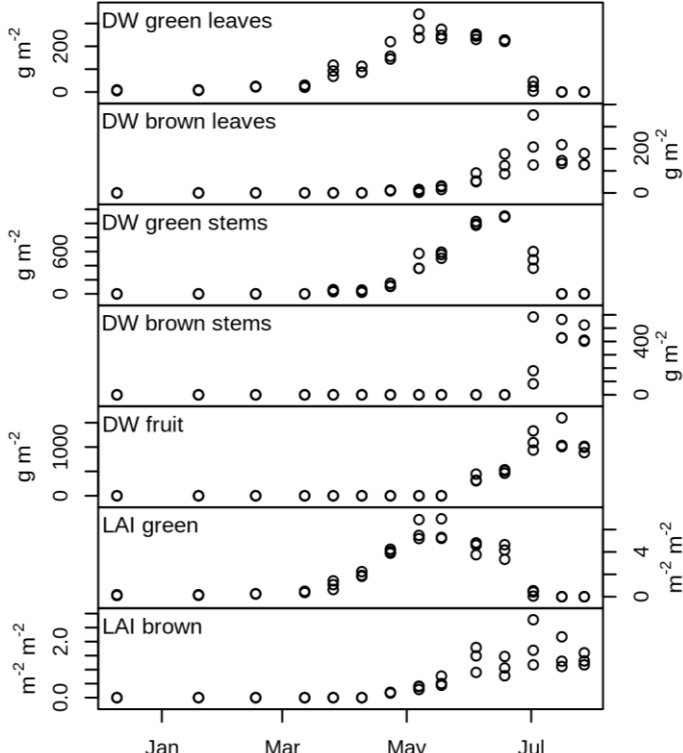

**Figure 3: Dry weight (DW) and leaf area index (LAI) of winter wheat on field F08 at the Selhausen site in 2009. Dataset identifier is SEF08WW09.**



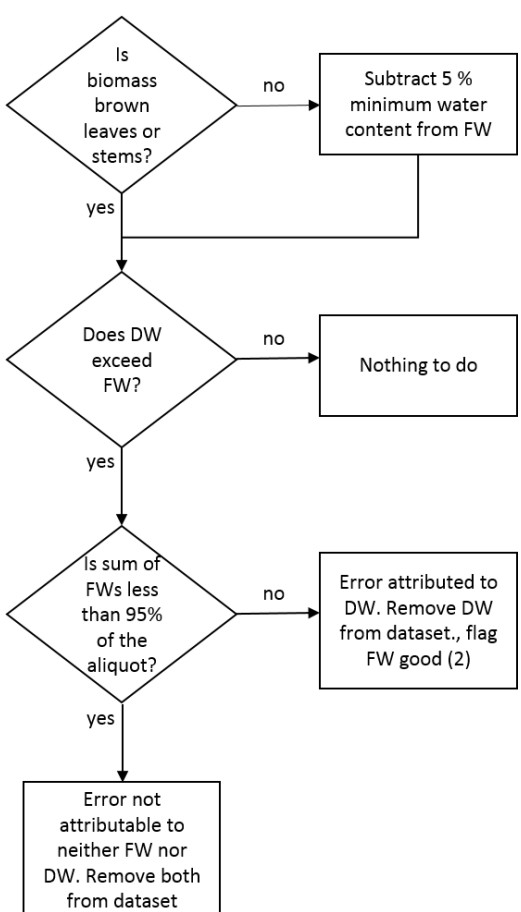

**Figure 4: Flow chart of the decision process of quality assurance when fresh weight (FW) was found larger than dry weight (DW).**



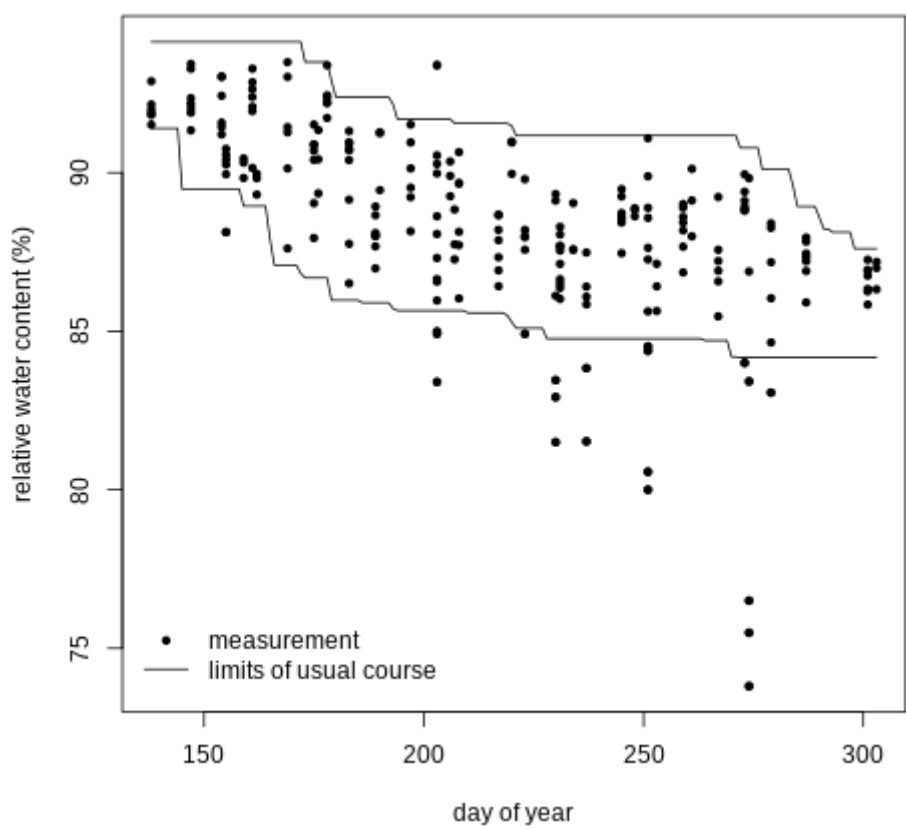

**Figure 5: Temporal course of sugar beet leaves' relative water content (percentage of fresh weight, all available data). Black lines show the upper and lower limit of the "usual course" (definition in the text). Blue circles denote meausrements outside the usual course and are therefore assigned the unusual water content flag (3). This data may still be valid because of heterogeneous conditions in a field (e.g. because of earlier drying).**





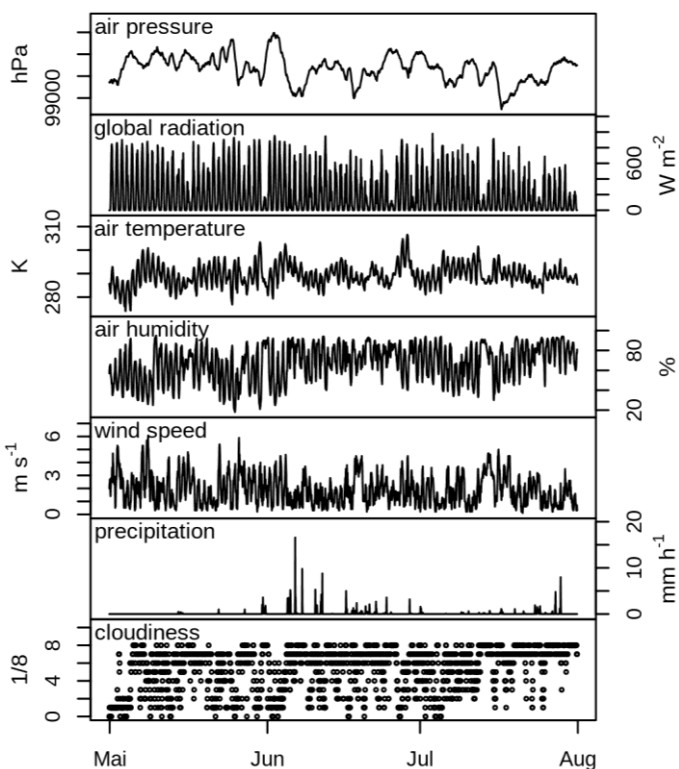

**Figure 6: Excerpt of the gap-filled hourly meteorological data for the Selhausen site for the period from May until July 2011.**





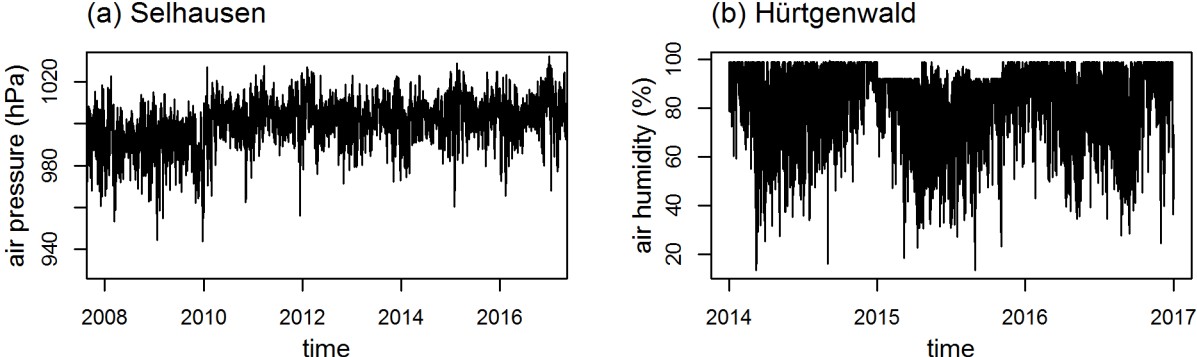

Figure 7: Breaks in time series of meteorological measurements. a) Air pressure in Selhausen, b) Air humidity in Hürtgenwald.





**Table 1: Terrain properties of the fields. Coordinates are for centroids, projection is UTM 32N (WGS 1984).**

| site | field | UTM N | UTM E | elev. (m) | area (ha) | slope (°) |
|---|---|---|---|---|---|---|
| Selhausen (SE) | F00 | 5638377 | 320584 | 105 | 1.2 | 1.5 |
| | F01 | 5638008 | 320341 | 103 | 9.7 | 0.4 |
| | F02 | 5637780 | 320428 | 104 | 7.4 | 0.4 |
| | F03 | 5638056 | 320643 | 105 | 4.0 | 1.1 |
| | F04 | 5638122 | 320826 | 109 | 1.9 | 0.7 |
| | F05 | 5637987 | 320860 | 110 | 2.4 | 0.9 |
| | F06 | 5637683 | 320723 | 107 | 2.4 | 1.5 |
| | F07 | 5638251 | 320613 | 105 | 2.4 | 1.4 |
| | F08 | 5638568 | 320538 | 104 | 6.5 | 1.4 |
| | F09 | 5638818 | 320403 | 102 | 2.6 | 0.8 |
| | F10 | 5638362 | 320408 | 103 | 2.2 | 0.7 |
| | F11 | 5638671 | 320699 | 106 | 4.8 | 1.0 |
| | F12 | 5638617 | 320713 | 107 | 3.1 | 0.7 |
| | F13 | 5638478 | 320742 | 108 | 0.8 | 0.4 |
| | F14 | 5638434 | 320754 | 109 | 0.6 | 0.5 |
| | F15 | 5638329 | 320600 | 105 | 0.7 | 1.9 |
| Merzenhausen (ME) | F01 | 5645502 | 310014 | 105 | 7.7 | 0.6 |
| Merken (MK) | F01 | 5636968 | 317442 | 93 | 0.7 | 0.7 |
| | F02 | 5635985 | 316781 | 108 | 5.3 | 0.6 |
| | F03 | 5636161 | 317011 | 116 | 6.1 | 0.4 |
| | F04 | 5635973 | 317223 | 114 | 4.3 | 0.4 |
| | F05 | 5635738 | 317217 | 115 | 1.1 | 0.5 |
| Hürtgenwald (HW) | F01 | 5622785 | 314460 | 360 | 8.4 | 2.4 |
| | F04 | 5621961 | 314387 | 373 | 6.8 | 1.1 |
| | F05 | 5621879 | 314156 | 374 | 5.7 | 2.6 |



**Table 2: Abbreviations for sites and land-use types**

| site | abbreviation |
| --- | --- |
| Selhausen | SE |
| Merken | MK |
| Merzenhausen | ME |
| Hürtgenwald | HW |

| land-use type | abbreviation |
| --- | --- |
| Catch crop | CC |
| Harvest residues* | HR |
| Maize | MA |
| Rapeseed | RA |
| Spelt | SP |
| Sugar beet | SB |
| Triticale | TC |
| Winter barley | WB |
| Winter wheat | WW |

\* period before sowing and after harvest independent of the actual presence of residues on the field



**Table 3: Data availability for vegetation data, fluxes, and management data ("X" data available, "-" no data available). For an explanation of vegetation data categories, refer to section 4. For crops, the year refers to harvest. Concerning vegetation data, number of points gives the maximum number of points in the field measured on the same date. In case of harvest residues, green sprouts and other biomass, data is only marked available if at least one value unequal zero is available.**

| Identifier | site | field | crop | variety / cultivar | year | preceding crop | EC data | sowing date | harvesting date | fertilization data | cultivation | sowing density | sowing depth | number of points (max) | number of dates | biomass per organ | aboveground biomass | LAI | C/N content | plant height | harvest residues (1=0) | Green sprouts (1=0) | Other (1=0) |
|---|---|---|---|---|---|---|---|---|---|---|---|---|---|---|---|---|---|---|---|---|---|---|---|
| SEF08WW08 | SE | F08 | WW | Raspail | 2008 | SB | X | X | X | X | - | - | - | 3 | 10 | X | - | X | X | X | - | - | X |
| SEF01SB08 | SE | F01 | SB | - | 2008 | - | - | X | - | X | - | X | - | 3 | 8 | X | - | X | X | X | - | - | - |
| SEF14MA08 | SE | F14 | MA | - | 2008 | - | - | X | - | X | - | - | - | 3 | 8 | X | - | X | X | X | - | - | X |
| SEF11RA08 | SE | F11 | RA | - | 2008 | WB | - | X | - | X | - | X | - | 3 | 9 | X | - | X | X | X | - | - | X |
| SEF08WW09 | SE | F08 | WW | Raspail | 2009 | WW | X | X | X | X | X | X | - | 3 | 14 | X | - | X | X | X | - | - | - |
| SEF07SB09 | SE | F07 | SB | Pauletta | 2009 | - | - | X | X | X | X | X | X | 3 | 11 | X | - | X | X | X | - | - | - |
| SEF10MA09 | SE | F10 | MA | Agro Lux | 2009 | - | - | X | X | X | X | X | X | 3 | 8 | X | - | X | X | X | - | - | - |
| SEF13RA09 | SE | F13 | RA | - | 2009 | - | - | X | X | X | X | - | X | 3 | 13 | X | - | X | X | X | - | - | X |
| SEF15WB09 | SE | F15 | WB | Laverda | 2009 | - | - | - | - | - | - | - | - | 3 | 8 | X | - | X | X | X | - | - | - |
| MKF05MA09 | MK | F05 | MA | Ronaldinho | 2009 | - | - | X | X | X | X | - | - | 3 | 7 | X | - | X | X | X | - | - | - |
| MKF01RA09 | MK | F01 | RA | NK-Fair | 2009 | - | - | X | X | X | X | X | X | 3 | 9 | X | - | X | X | X | - | - | - |
| MKF04SB09 | MK | F04 | SB | Beretta KWS | 2009 | - | X* | X | X | X | X | X | X | 3 | 10 | X | - | X | X | X | - | - | - |
| MKF03WB09 | MK | F03 | WB | Fridericus | 2009 | - | X* | X | X | X | - | X | X | 3 | 8 | X | - | X | X | X | - | - | - |
| MKF02WW09 | MK | F02 | WW | Hattrick | 2009 | - | X* | X | X | X | X | - | X | 3 | 10 | X | - | X | X | X | - | - | - |
| SEF07WW10 | SE | F07 | WW | - | 2010 | SB | - | - | - | - | - | - | - | 6 | 10 | X | - | X | X | X | - | - | - |
| SEF08SB10 | SE | F08 | SB | Supero | 2010 | WW | - | X | X | X | X | - | X | 7 | 12 | X | - | X | X | X | - | - | - |
| SEF09MA10 | SE | F09 | MA | - | 2010 | - | - | - | - | - | - | - | - | 3 | 1 | X | - | X | - | X | - | - | - |
| SEF12RA10 | SE | F12 | RA | - | 2010 | - | - | - | - | - | - | - | - | 5 | 8 | X | - | X | X | X | - | - | - |



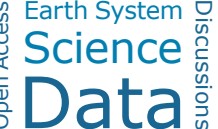

| | | | | | | | | | | | | | | | | | | | | | | | |
|---|---|---|---|---|---|---|---|---|---|---|---|---|---|---|---|---|---|---|---|---|---|---|---|
| SEF01WB10 | SE | F01 | WB | - | 2010 | - | - | - | - | - | - | - | - | 5 | 8 | X | - | X | - | X | - | - | - |
| MKF04WW10 | MK | F04 | WW | - | 2010 | SB | - | - | - | - | - | - | - | 3 | 2 | X | - | X | - | X | - | - | - |
| MKF02WB10 | MK | F02 | WB | - | 2010 | WW | - | - | - | - | - | - | - | 3 | 2 | X | - | X | - | X | - | - | - |
| MEF01HR11 | ME | F01 | HR | - | 2011 | - | X | - | - | - | - | - | - | 12 | 1 | - | - | - | X | - | X | - | - |
| MEF01WW11 | ME | F01 | WW | Potenzial | 2011 | SB | X | X | X | X | - | X | X | 12 | 10 | X | - | X | X | X | - | - | - |
| MEF01HR12 | ME | F01 | HR | - | 2012 | - | X | - | - | - | - | - | - | 12 | 1 | - | - | - | - | - | X | - | - |
| MEF01WW12 | ME | F01 | WW | Tobak | 2012 | WW | X | X | X | X | - | X | X | 12 | 12 | X | - | X | X | X | - | - | - |
| SEF04WW13 | SE | F04 | WW | - | 2013 | - | - | - | - | - | - | - | - | 6 | 10 | X | - | X | X | X | - | - | - |
| SEF01HR15 | SE | F01 | HR | - | 2015 | - | X | - | - | - | - | - | - | 3 | 3 | - | - | - | - | X | X | - | - |
| SEF01WW15 | SE | F01 | WW | Premio | 2015 | - | X | X | X | X | - | - | - | 3 | 6 | X | - | X | - | X | - | - | - |
| SEF03HR15 | SE | F03 | HR | - | 2015 | - | - | - | - | - | - | - | - | 3 | 5 | - | - | - | - | - | X | X | X |
| SEF03WW15 | SE | F03 | WW | - | 2015 | - | - | X | X | X | - | X | X | 3 | 6 | X | X | X | - | X | - | - | - |
| SEF02SB15 | SE | F02 | SB | - | 2015 | - | - | - | - | - | - | - | - | 3 | 7 | X | - | X | - | X | - | - | X |
| SEF04HR15 | SE | F04 | HR | - | 2015 | - | - | - | - | - | - | - | - | 3 | 5 | - | - | - | - | - | X | X | X |
| SEF04SP15 | SE | F04 | SP | - | 2015 | WW | - | - | - | - | - | - | - | 3 | 4 | X | - | X | - | X | - | - | - |
| SEF05HR15 | SE | F05 | HR | - | 2015 | - | - | - | - | - | - | - | - | 3 | 3 | - | - | - | - | - | X | X | - |
| SEF05WW15 | SE | F05 | WW | - | 2015 | - | - | - | - | - | - | - | - | 3 | 4 | X | X | X | - | X | - | - | - |
| HWF01HR15 | HW | F01 | HR | - | 2015 | - | - | - | - | - | - | - | - | 3 | 2 | - | - | - | - | - | X | X | - |
| HWF01MA15 | HW | F01 | MA | Silage maize | 2015 | - | - | X | X | X | X | X | X | 3 | 7 | X | - | X | - | X | - | - | X |
| HWF04HR15 | HW | F04 | HR | - | 2015 | - | - | - | - | - | - | - | - | 0 | 1 | - | - | - | - | - | - | - | - |
| HWF04TC15 | HW | F04 | TC | Winter TC | 2015 | - | - | X | X | X | X | - | X | 3 | 3 | X | X | X | - | X | - | - | X |
| SEF01WB16 | SE | F01 | WB | - | 2016 | WW | X | X | X | X | X | - | - | 3 | 7 | X | - | X | - | X | - | - | - |
| SEF01HR16 | SE | F01 | HR | - | 2016 | - | X | - | - | - | - | - | - | 3 | 3 | - | - | - | - | X | X | X | - |
| SEF01CC16 | SE | F01 | CC | - | 2016 | WB | X | X | X | X | X | - | - | 3 | 2 | - | X | - | - | X | - | - | - |
| SEF03HR16 | SE | F03 | HR | - | 2016 | WW | - | - | - | - | - | - | - | 3 | 3 | - | - | - | - | - | X | - | - |
| SEF03SB16 | SE | F03 | SB | - | 2016 | WW | - | - | - | - | - | - | - | 3 | 7 | X | - | X | - | X | - | - | - |
| SEF04HR16 | SE | F04 | HR | - | 2016 | - | - | - | - | - | - | - | - | 0 | 2 | - | - | - | - | - | - | - | - |

| | | | | | | | | | | | | | | | | | | | | | | | |
|---|---|---|---|---|---|---|---|---|---|---|---|---|---|---|---|---|---|---|---|---|---|---|---|
| SEF04SB16 | SE | F04 | SB | Kleist | 2016 | SP | - | X | - | X | X | X | X | 3 | 7 | X | - | X | - | X | - | - | - |
| SEF05HR16 | SE | F05 | HR | - | 2016 | WW | - | - | - | - | - | - | - | 3 | 3 | - | - | - | - | X | X | - | - |
| SEF05WW16 | SE | F05 | WW | - | 2016 | WW | - | - | - | - | - | - | - | 3 | 7 | X | X | X | - | X | - | - | - |
| SEF06HR16 | SE | F06 | HR | - | 2016 | - | - | - | - | - | - | - | - | 4 | 2 | - | - | - | - | X | X | - | - |
| SEF06WB16 | SE | F06 | WB | - | 2016 | - | - | - | - | - | - | - | - | 4 | 7 | X | - | X | - | X | - | - | - |
| HWF01HR16 | HW | F01 | HR | - | 2016 | - | - | - | - | - | - | - | - | 3 | 4 | - | - | - | - | X | X | - | - |
| HWF01MA16 | HW | F01 | MA | - | 2016 | MA | - | - | X | X | X | X | X | 3 | 4 | X | - | X | - | X | - | - | X |
| HWF05HR16 | HW | F05 | HR | - | 2016 | - | - | - | - | - | - | - | - | 0 | 1 | - | - | - | - | X | - | - | - |
| HWF05TC16 | HW | F05 | TC | - | 2016 | - | - | X | X | - | X | - | X | 3 | 4 | X | X | X | - | X | X | - | - |
| SEF06WB17 | SE | F06 | WB | - | 2017 | WB | - | - | - | - | - | - | - | 6 | 1 | X | - | X | - | X | - | - | - |
| MEF01HR17 | ME | F01 | HR | - | 2017 | - | X | - | - | - | - | - | - | 3 | 1 | - | - | - | - | X | - | - | - |
| MEF01WW17 | ME | F01 | WW | - | 2017 | - | X | X | - | - | X | - | - | 3 | 7 | X | - | X | - | X | - | - | X |

* Data from two heights



**Table 4: Quality flags set in case the sum of its components differed from the aliquot.**

|              | < 5 %            | 5 %–10 %          | 10 %–15 %         | > 15 %           |
|--------------|------------------|-------------------|-------------------|------------------|
| sum < aliquot | high quality (1) | good quality (2)  | suspicious (4)    | low quality (5)  |
| sum > aliquot | high quality (1) | suspicious (4)    | suspicious (4)    | low quality (5)  |





**Table 5: Locations, processing software and instrument heights of eddy covariance stations. Coordinates are UTM zone 32N (WGS1984). For information on quality indicators see section 5.2.**

| site | field | year | Identifier | UTM N | UTM E | elevation (m a.s.l.) | processing software | quality indicator | height (cm)* |
|------|-------|------|------------|-------|-------|----------|----------|----------|--------|
| SE | F08 | 2007 | SEF08_SE_EC000_fluxes_2007 | 5638560 | 320543 | 103 | TK2 | flags | 245 |
| SE | F08 | 2008 | SEF08_SE_EC000_fluxes_2008 | 5638560 | 320543 | 103 | TK2 | flags | 245 |
| SE | F08 | 2009 | SEF08_SE_EC000_fluxes_2009 | 5638560 | 320543 | 103 | TK2 | flags | 245 |
| SE | F08 | 2010 | SEF08_SE_EC000_fluxes_2010 | 5638560 | 320543 | 103 | TK2 | flags | 245 |
| SE | F01 | 2015 | SEF01_SE_EC001_fluxes_2015 | 5638010 | 320380 | 103 | TK3.1 | flags | 245 |
| SE | F01 | 2016 | SEF01_SE_EC001_fluxes_2016 | 5638010 | 320380 | 103 | TK3.1 | flags | 245 |
| ME | F01 | 2011 | MEF01_ME_EC001_fluxes_2011 | 5645497 | 310059 | 93 | TK3.1 | flags | 198 |
| ME | F01 | 2012 | MEF01_ME_EC001_fluxes_2012 | 5645497 | 310059 | 93 | TK3.1 | flags | 198 |
| ME | F01 | 2017 | MEF01_ME_EC001_fluxes_2017 | 5645497 | 310059 | 93 | TK3.1 | flags | 198 |
| MK | F02 | 2009 | MKF02_MK_ECJ1l_fluxes_2009 | 5635998 | 316798 | 116 | ECpack 2.5.20 | tolerances | 240 |
| MK | F02 | 2009 | MKF02_MK_ECJ1u_fluxes_2009 | 5635998 | 316798 | 116 | ECpack 2.5.20 | tolerances | 590 |
| MK | F03 | 2009 | MKF03_MK_ECS4l_fluxes_2009 | 5636165 | 317010 | 114 | ECpack 2.5.20 | tolerances | 260 |
| MK | F03 | 2009 | MKF03_MK_ECS4u_fluxes_2009 | 5636165 | 317010 | 114 | ECpack 2.5.20 | tolerances | 596 |
| MK | F04 | 2009 | MKF04_MK_ECS3l_fluxes_2009 | 5635956 | 317204 | 115 | ECpack 2.5.20 | tolerances | 248 |
| MK | F04 | 2009 | MKF04_MK_ECS3u_fluxes_2009 | 5635956 | 317204 | 115 | ECpack 2.5.20 | tolerances | 604 |

*height: instrument height of anemometer and IRGA (above ground)



**Table 6: Flag values set by the TK software and their meanings.**

| flag | meaning |
| --- | --- |
| 0 | high quality data, use in fundamental research possible |
| 1 | moderate quality data, no restrictions for use in long term observation programs |
| 2 | low data quality, gap filling necessary |



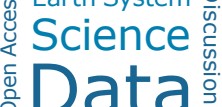

**Table 7: Acceptable value ranges of ECpack results and tolerance-values at the lower and upper boundary. For the meaning of the group-column refer to the text.**

| variable | lower boundary | tolerance at lower boundary | upper boundary | tolerance at upper boundary | group |
|---|---|---|---|---|---|
| Mean(u) | 0 | 0.2 | 200 | 1 | A |
| Mean(w) | 0 | 0.05 | 20 | 0.1 | A |
| Mean(TSon) | 273.15 | 0.1 | 350 | 0.1 | A |
| Mean(rhoV) | 0 | 2E-4 | 0.2 | 2E-4 | B |
| Mean(qCO2) | 4E-4 | 1E-5 | 1E-3 | 1E-5 | B |
| U_dir | 0 | 180 | 360 | 180 | A |
| Hsonic | 0 | 25 | 1000 | 100 | A |
| SumLvE | 0 | 50 | 1000 | 200 | B |
| Ustar | 0 | 0.2 | 1.0 | 0.3 | A |
| SumFCO2 | 0 | 5E-7 | 5E-6 | 1E-6 | B |



**Table 8: Availability of soil information per site. C/N: carbon and nitrogen content are not always both available. Due to the absence of carbonates, C-content is expected to equal SOC.**

| Site | particle sizes fine soil | gravel | bulk density | SOC | C and/or N |
|------|--------------------------|--------|--------------|-----|------------|
| HW | X | X | X | - | - |
| MK | X | - | - | - | X |
| ME | X | X | X | X | X |
| SE | X | X | X | - | X |

**Table 9: Meteorological Stations, their positions, available data and data source**

| ID | station | UTM northing | UTM easting | period used | AirHum | AirPres | AirTemp | Precip | Globrad | Wind | Cloud | data source |
|---|---|---|---|---|---|---|---|---|---|---|---|---|
| 1 | ME_BCK_001 | 5645555 | 310095 | 2011–2016 | X | X | | X | X | X | | TEODOOR[1] |
| 2 | RO_AKRW_003 | 5611891 | 309102 | 2011–2016 | X | X | X | X | | X | | TEODOOR[1] |
| 3 | RO_BKY_010 | 5611219 | 309322 | 2012–2016 | X | X | X | X | X | X | | TEODOOR[1] |
| 4 | RO_EC_001 | 5611250 | 309312 | 2011–2016 | X | X | X | X | X | X | | TEODOOR[1] |
| 5 | RU_BCK_002 | 5652036 | 312165 | 2011–2016 | X | X | X | X | X | X | | TEODOOR[1] |
| 6 | RU_BCK_003 | 5637669 | 318956 | 2011–2016 | X | X | X | X | | X | | TEODOOR[1] |
| 7 | RU_BCK_004 | 5668397 | 301947 | 2012–2016 | X | X | X | X | | X | | TEODOOR[1] |
| 8 | RU_BCDKR_001 | 5599172 | 313945 | 2011–2016 | X | X | X | X | X | X | | TEODOOR[1] |
| 9 | RU_K_002 | 5642873 | 317452 | 2013–2016 | X | X | X | X | X | X | | TEODOOR[1] |
| 10 | SE_BDK_999 | 5638335 | 320536 | 2009–2016 | X | X | X | X | X | X | | TEODOOR[1,3] |
| 11 | SE_EC_001 | 5638012 | 320375 | 2011–2016 | X | X | X | X | X | X | | TEODOOR[1] |
| 12 | WU_BKY_010 | 5597950 | 310540 | 2012–2016 | X | X | X | X | X | X | | TEODOOR[1] |
| 13 | WU_EC_002 | 5597955 | 311089 | 2013–2016 | X | X | X | | X | X | | TEODOOR[1] |
| 14 | WU_K_002 | 5597960 | 311091 | 2014–2016 | X | X | X | X | | X | | TEODOOR[1] |
| 15 | ME_EC_001 | 5645497 | 310059 | 2011–2016 | X | X | X | X | X | X | | TEODOOR[1] |
| 16 | RU_K_001 | 5643013 | 317883 | 2007–2016 | X | X | | X | X | X | | TEODOOR[1] |
| 17 | RU_EC_001 | 5637813 | 318969 | 2011–2016 | X | X | X | | X | X | | TEODOOR[1] |
| 18 | WU_EC_001 | 5598173 | 310739 | 2011–2014 | X | X | X | | X | X | | TEODOOR[1] |
| 19 | SE_EC_002 | 5638375 | 320591 | 2010 | X | X | X | X | X | X | | TEODOOR[1] |
| 20 | HW_BK_001 | 5622292 | 314567 | 2015–2016 | X | X | X | X | X | X | | GLOBE[4] |
| 21 | HW_BK_002 | 5621923 | 314600 | 2016 | X | X | X | X | X | X | | GLOBE[4,5] |
| 22 | 10501 | 5629698 | 295161 | 2007–2010 | | X | | | | | X | DWD[6,7] |
| 23 | 10505 | 5631617 | 290318 | 2011–2016 | | | | | | | X | DWD[6,8] |
| 24 | H827 | 5616739 | 317991 | 2014–2015 | X | | X | X | | X | | DWD[6] |
| 25 | SE_EC_000 | 5638537 | 320558 | 2007–2009 | X | X | X | X | X | X | | TR32DB[2] |

[1] http://teodoor.icg.kfa-juelich.de/ibg3searchportal2/, Eifel/Lower Rhine Valley Observatory
[2] http://www.tr32db.de
[3] includes data from stations SE_BK_001 and SE_BDK_002 from TEODOOR[1]
[4] https://datasearch.globe.gov/
[5] consists of three stations





---

[6] Deutscher Wetterdienst, DWD, ftp://ftp-cdc.dwd.de/pub/CDC/observations_germany/climate/hourly/

[7] DWD station Aachen, old location

[8] DWD station Aachen, new location



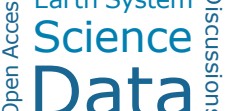

**Table 10: Source of meteorological data given as station IDs as defined in Table 9. Station IDs in parenthesis are stations used for gap-filling. Colum "Met" shows the method used for gap-filling as explained in the text.**

| year | AirPres | GlobRad | AirTemp | AirHum | Wind | Precip | Cloud | Met |
|---|---|---|---|---|---|---|---|---|
| Hürtgenwald (HW) | | | | | | | | |
| 2014 | 4 (24) | 4 (24) | 4 (24) | 4 (24) | 4 (24) | 4 (24) | 23 | 1 |
| 2015 | 20 (4, 24) | 20 (4, 24) | 20 (4, 24) | 20 (4, 24) | 20 (4, 24) | 20 (4, 24) | 23 | 1 |
| 2016 | 20 (4) | 20 (21) | 20 (21) | 20 (21) | 20 (21) | 20 (21) | 23 | 1 |
| Selhausen (SE) | | | | | | | | |
| 2007 | 25 (22) | 22 | 25 (16) | 25 (16) | 25 (16) | 16 | 22 | 0 |
| 2008-2009 | 25 (22) | 22 | 25 (16) | 25 (16) | 25 (10,16) | 16 | 22 | 0 |
| 2010 | 22 | 11 (19, 10, 16) | 11 (19, 10, 16) | 10 (16) | 11 (19, 10, 16) | 16 | 22 | 2 |
| 2011-2017 | 10 (1–18) | 11 (1–18) | 10 (1–18) | 10 (1–18) | 11 (1–18) | 10 (1–18) | 23 | 3 |
| Merzenhausen (ME) | | | | | | | | |
| 2009-2010 | 22 | 15 (1, 16) | 15 (1, 16) | 15 (1, 16) | 15 (1, 16) | 15 (1, 16) | 22 | 2 |
| 2011-2017 | 1 (2–18) | 15 (1–18) | 1 (2–18) | 1 (2–18) | 1 (2–18) | 1 (2–18) | 23 | 3 |



**Table 11: Instrument uncertainties for meterological measurements at stations 1 to 21 and 25 (for station IDs compare Table 9).**

| variable | Uncertainty |
|---|---|
| Air pressure (Pa) | p_ref (flux data): 1 % for relative humidity<br>AirPres (weather data): 0 to 30°C: ±0.5 hPa; -52 to +60°C: ±1 hPa |
| global radiation (W m$^{-2}$) | ±5 % to ±10 % for daily sums |
| Air temperature (°C) | ±0.2 – 0.4°C |
| Air humidity (%) | Accuracy at 20°C:<br>±2 % RH (0 to 90 % Relative Humidity); ±3 % RH (90 to 100 % Relative Humidity) |
| Wind speed (m s$^{-1}$) | Offset error:<br>< ±8.0 cm s$^{-1}$ (u, v), < ±4.0 cm s$^{-1}$ (z)<br>Gain Error:<br>Wind Vector within ±5° of horizontal: <±2% of reading<br>Wind Vector within ±10° of horizontal: <±3% of reading<br>Wind Vector within ±20° of horizontal: <±6% of reading |
| Precipitation (mm h$^{-1}$) | < 3 % |
| Cloudiness (1/8) | Uncertainty unknown |
| Wind direction (°)* | ±0.7° at 1 m s$^{-1}$ for horizontal wind |

* included in flux datafiles



**Table 12: Uncertainty estimates for weather timeseries gap-filling expressed as pairs of bias (B) and RMSE (R). Missing data is denoted by NA; "-" marks cases where there were no gaps or no data. Methods are described in the text. For abbreviations see Table A7.**

| Period | AirPres (Pa) | | GlobRad (W m$^{-2}$) | | AirTemp (K) | | AirHum (%) | | Wind (m s$^{-1}$) | | Precip (mm) | | InLW (W m$^{-2}$) | |
|---|---|---|---|---|---|---|---|---|---|---|---|---|---|---|
| | B | R | B | R | B | R | B | R | B | R | B | R | B | R |
| 2007 to 2009 | 24.7 | 27.2 | - | - | 0.1 | 0.9 | -5.7 | 9.5 | -0.4 | 0.9 | NA | NA | - | - |
| 2010 | - | - | NA | NA | -0.3 | 0.8 | -5.7 | 8.0 | 0.1 | 0.8 | NA | NA | - | - |
| 2011 to 2017 | -7.6 | 12.2 | 0.6 | 78.8 | 0.1 | 0.9 | 0.3 | 1.2 | -1.6 | 2.5 | -0.1 | 0.3 | -5.8 | 24.6 |



**Table A1: Columns in the vegetation data files**

| #Col. | variable | units | data type | description |
|---|---|---|---|---|
| 1 | dataset | - | character | Dataset name |
| 2 | test_site | - | character | Site ID |
| 3 | field | - | character | Field ID (per site) |
| 4 | land_use | - | character | Land-use ID |
| 5 | Date | - | date | Date of field measurement (YYYY-MM-DD) |
| 6 | time | - | time | UTC-time of field measurement (hh:mm) |
| 7 | UTM_northing | m | numeric | UTM Northing (WGS84, 32N) |
| 8 | UTM_northing_FLAG | - | numeric | UTM Northing quality flag |
| 9 | UTM_easting | m | numeric | UTM Easting (WGS84, 32N) |
| 10 | UTM_easting_FLAG | - | numeric | UTM Easting quality flag |
| 11 | canopy_height | cm | numeric | Height of the canopy |
| 12 | bbch | BBCH* | character | Phenological development state (BBCH scale) |
| 13 | num_plants_m2 | $m^{-2}$ | numeric | Number of plants per m^2 (calculated) |
| 14 | LAI_green | $m^2\ m^{-2}$ | numeric | Green LAI |
| 15 | LAI_green_FLAG | - | numeric | Green LAI quality flag |
| 16 | LAI_brown | $m^2\ m^{-2}$ | numeric | Brown LAI |
| 17 | LAI_brown_FLAG | - | numeric | Brown LAI quality flag |
| 18 | FW_green_leaves | $g\ m^{-2}$ | numeric | Fresh weight of green leaves |
| 19 | FW_green_leaves_FLAG | - | numeric | Fresh weight of green leaves quality flag |
| 20 | DW_green_leaves | $g\ m^{-2}$ | numeric | Dry weight of green leaves |
| 21 | DW_green_leaves_FLAG | - | numeric | Dry weight of green leaves quality flag |
| 22 | FW_brown_leaves | $g\ m^{-2}$ | numeric | Fresh weight of brown leaves |
| 23 | FW_brown_leaves_FLAG | - | numeric | Fresh weight of brown leaves quality flag |
| 24 | DW_brown_leaves | $g\ m^{-2}$ | numeric | Dry weight of brown leaves |
| 25 | DW_brown_leaves_FLAG | - | numeric | Dry weight of brown leaves quality flag |
| 26 | FW_green_stems | $g\ m^{-2}$ | numeric | Fresh weight of green stems/tillers/stalks |
| 27 | FW_green_stems_FLAG | - | numeric | Fresh weight of green stems/tillers/stalks quality flag |
| 28 | DW_green_stems | $g\ m^{-2}$ | numeric | Dry weight of green stems/tillers/stalks |



| 29 | DW_green_stems_FLAG | - | numeric | Dry weight of green stems/tillers/stalks quality flag |
|---|---|---|---|---|
| 30 | FW_brown_stems | g m$^{-2}$ | numeric | Fresh weight of brown stems/tillers/stalks |
| 31 | FW_brown_stems_FLAG | - | numeric | Fresh weight of brown stems/tillers/stalks quality flag |
| 32 | DW_brown_stems | g m$^{-2}$ | numeric | Dry weight of brown stems/tillers/stalks |
| 33 | DW_brown_stems_FLAG | - | numeric | Dry weight of brown stems/tillers/stalks quality flag |
| 34 | FW_fruit | g m$^{-2}$ | numeric | Fresh weight of harvest organ (e.g. fruit, beet) |
| 35 | FW_fruit_FLAG | - | numeric | Fresh weight of harvest organ quality flag |
| 36 | DW_fruit | g m$^{-2}$ | numeric | Dry weight of harvest organ (e.g. fruit, beet) |
| 37 | DW_fruit_FLAG | - | numeric | Dry weight of harvest organ quality flag |
| 38 | FW_biomass_undiff | g m$^{-2}$ | numeric | Fresh weight of aboveground biomass not separated into organs |
| 39 | FW_biomass_undiff_FLAG | - | numeric | Fresh weight of aboveground biomass not separated into organs quality flag |
| 40 | DW_biomass_undiff | g m$^{-2}$ | numeric | Dry weight of aboveground biomass not separated into organs |
| 41 | DW_biomass_undiff_FLAG | - | numeric | Dry weight of aboveground biomass not separated into organs quality flag |
| 42 | FW_harvest_residues | g m$^{-2}$ | numeric | Fresh weight of harvest residues |
| 43 | FW_harvest_residues_FLAG | - | numeric | Fresh weight of harvest residues quality flag |
| 44 | DW_harvest_residues | g m$^{-2}$ | numeric | Dry weight of harvest residues |
| 45 | DW_harvest_residues_FLAG | - | numeric | Dry weight of harvest residues quality flag |
| 46 | FW_green_sprouts | g m$^{-2}$ | numeric | Fresh weight of green sprouts (growing between harvest residues) |
| 47 | FW_green_sprouts_FLAG | - | numeric | Fresh weight of green sprouts quality flag |
| 48 | DW_green_sprouts | g m$^{-2}$ | numeric | Dry weight of green sprouts (growing between harvest residues) |
| 49 | DW_green_sprouts_FLAG | - | numeric | Dry weight of green sprouts quality flag |



| | | | | |
|---|---|---|---|---|
| 50 | FW_other | g m⁻² | numeric | Fresh weight of other biomass (e.g. weeds) |
| 51 | FW_other_FLAG | - | numeric | Fresh weight of other biomass quality flag |
| 52 | DW_other | g m⁻² | numeric | Dry weight of other biomass (e.g. weeds) |
| 53 | DW_other_FLAG | - | numeric | Dry weight of other biomass quality flag |
| 54 | other_descr | - | character | type of biomass measured as "biomass_other" |
| 55 | N_green_leaves | mass% | numeric | Relative nitrogen content of green leaves |
| 56 | C_green_leaves | mass% | numeric | Relative carbon content of green leaves |
| 57 | N_brown_leaves | mass% | numeric | Relative nitrogen content of brown leaves |
| 58 | C_brown_leaves | mass% | numeric | Relative carbon content of brown leaves |
| 59 | N_green_stems | mass% | numeric | Relative nitrogen content of green stems/tillers/stalks |
| 60 | C_green_stems | mass% | numeric | Relative carbon content of green stems/tillers/stalks |
| 61 | N_brown_stems | mass% | numeric | Relative nitrogen content of brown stems/tillers/stalks |
| 62 | C_brown_stems | mass% | numeric | Relative carbon content of brown stems/tillers/stalks |
| 63 | N_fruit | mass% | numeric | Relative nitrogen content of harvest organ (e.g. fruit, beet) |
| 64 | C_fruit | mass% | numeric | Relative carbon content of harvest organ (e.g. fruit, beet) |
| 65 | N_biomass_undiff | mass% | numeric | Relative nitrogen content of aboveground biomass not separated into organs |
| 66 | C_biomass_undiff | mass% | numeric | Relative carbon content of aboveground biomass not separated into organs |
| 67 | N_harvest_residues | mass% | numeric | Relative nitrogen content of harvest residues |
| 68 | C_harvest_residues | mass% | numeric | Relative carbon content of harvest residues |
| 69 | N_green_sprouts | mass% | numeric | Relative nitrogen content of green sprouts (growing between harvest residues) |



| 70 | C_green_sprouts | mass% | numeric | Relative carbon content of green sprouts (growing between harvest residues) |
| 71 | N_other | mass% | numeric | Relative nitrogen content of other biomass (e.g. weeds) |
| 72 | C_other | mass% | numeric | Relative carbon content of other biomass (e.g. weeds) |
| 73 | is_cn_field_mean | - | logical | Have C- and N-contents been measured from a composite sampled from all points in the field? |
| 74 | soil_unit | - | character | Assignment to a soil unit of Brogi et al. (2019), only Selhausen |
| 75 | comment | - | character | Comment |

* see Meier et al. (2009) and references therein



**Table A2: Columns of flux datafiles processed with the software ECpack. With the exception of the timestamps, all datatypes are numeric.**

| #Col. | variable | Units | description |
|---|---|---|---|
| 1 | Datetime(end) | - | UTC-time end of interval (YYYY-MM-DD hh:mm) |
| 2 | #Samples | - | Number of records aggregated to data in the current row |
| 3 | Mean(u) | m s$^{-1}$ | horizontal wind component (coordinate system turned into mean wind) |
| 4 | TolMean(u) | m s$^{-1}$ | Estimate of 95% confidence intervals for horizontal wind component u |
| 5 | Mean(v) | m s$^{-1}$ | horizontal wind component orthogonal to v (almost zero due to rotation of coordinate system) |
| 6 | TolMean(v) | m s$^{-1}$ | Estimate of 95% confidence intervals for horizontal wind component v |
| 7 | Mean(w) | m s$^{-1}$ | vertical wind (after planar fit rotation) |
| 8 | TolMean(w) | m s$^{-1}$ | Estimate of 95% confidence intervals for vertical wind speed |
| 9 | Mean(TSon) | K | air temperature, calculated from sonic temperature, pressure and $H_2O$ density |
| 10 | TolMean(TSon) | K | Estimate of 95% confidence intervals for air temperature |
| 11 | Mean(rhoV) | kg m$^{-3}$ | $H_2O$ density |
| 12 | TolMean(rhoV) | kg m$^{-3}$ | Estimate of 95% confidence intervals for average $H_2O$ density |
| 13 | Mean(qCO2) | kg kg$^{-1}$ | $CO_2$ mixing ratio |
| 14 | TolMean(qCO2) | kg kg$^{-1}$ | Estimate of 95% confidence intervals for average $CO_2$ mixing ratio |
| 15 | Std(u) | m s$^{-1}$ | Standard deviation of horizontal wind component u |
| 16 | TolStd(u) | m s$^{-1}$ | Estimate of 95% confidence intervals for horizontal wind component u |
| 17 | Std(v) | m s$^{-1}$ | Standard deviation of horizontal wind component v |
| 18 | TolStd(v) | m s$^{-1}$ | Estimate of 95% confidence intervals for horizontal wind component v |
| 19 | Std(w) | m s$^{-1}$ | Standard deviation of vertical wind speed |
| 20 | TolStd(w) | m s$^{-1}$ | Estimate of 95% confidence intervals for vertical wind speed |
| 21 | Std(TSon) | K | Standard deviation of sonic temperature |





| 22 | TolStd(TSon) | K | Estimate of 95% confidence intervals for standard deviation of air temperature |
|---|---|---|---|
| 23 | Std(q) | kg kg$^{-1}$ | Standard deviation of specific humidity |
| 24 | TolStd(q) | kg kg$^{-1}$ | Estimate of 95% confidence intervals for specific humidity |
| 25 | Std(qCO2) | kg kg$^{-1}$ | Standard deviation of $CO_2$ mixing ratio |
| 26 | TolStd(qCO2) | kg kg$^{-1}$ | Estimate of 95% confidence intervals for standard deviation of average $CO_2$ mixing ratio |
| 27 | Cov(u*v) | m$^2$ s$^{-2}$ | Covariance of wind components u and v |
| 28 | TolCov(u*v) | m$^2$ s$^{-2}$ | Estimate of 95% confidence intervals for covariance of wind components u and v |
| 29 | Cov(v*w) | m$^2$ s$^{-2}$ | Covariance of wind components v and w |
| 30 | TolCov(u*w) | m$^2$ s$^{-2}$ | Estimate of 95% confidence intervals for covariance of wind components u and w |
| 31 | Cov(u*w) | m$^2$ s$^{-2}$ | Covariance of wind components u and w |
| 32 | TolCov(v*w) | m$^2$ s$^{-2}$ | Estimate of 95% confidence intervals for covariance of wind components v and w |
| 33 | RhoSon | kg m$^{-3}$ | Air density from the ultrasonic anemometer |
| 34 | Tol(RhoSon) | kg m$^{-3}$ | Estimate of 95% confidence intervals for air density from the ultrasonic anemometer |
| 35 | U_vect | m s$^{-1}$ | In this processing scheme, identical to Mean(u) |
| 36 | Tol(U_vect) | m s$^{-1}$ | Estimate of 95% confidence intervals for U_vect |
| 37 | U_dir | ° | Wind direction in geographical coordinate system |
| 38 | Tol(U_dir) | ° | Estimate of 95% confidence intervals for wind direction in geographical coordinate system |
| 39 | HSonic | W/m² | Sensible heat flux including planar fit, Moore and Schotanus correction |
| 40 | Tol(HSonic) | W m$^{-2}$ | Estimate of 95% confidence intervals for sensible heat flux |
| 41 | SumLvE | W m$^{-2}$ | Latent heat flux including planar fit, Moore and WPL correction |
| 42 | Tol(SumLvE) | W m$^{-2}$ | Estimate of 95% confidence intervals for latent heat flux |
| 43 | Ustar | m s$^{-1}$ | Friction velocity including planar fit and Moore correction |
| 44 | Tol(Ustar) | m s$^{-1}$ | Estimate of 95% confidence intervals for friction velocity |
| 45 | SumFCO2 | kg m$^{-2}$ s$^{-1}$ | $CO_2$ flux without consideration of storage flux |



| 46 | Tol(SumFCO2) | kg m$^{-2}$ s$^{-1}$ | Estimate of 95% confidence intervals for $CO_2$ flux without consideration of storage flux |
|----|--------------|----------------------|---------------------------------------------------------------------------------------------|



**Table A3: Columns of flux datafiles processed with the software TK. With the exception of the timestamps, all datatypes are numeric.**

| #Col. | variable | units | Description |
|---|---|---|---|
| 1 | T_begin | - | UTC-time beginning of interval (YYYY-MM-DD hh:mm) |
| 2 | T_end | - | UTC-time end of interval (YYYY-MM-DD hh:mm) |
| 3 | u | $m\ s^{-1}$ | Horizontal wind speed (coordinate system turned into mean wind) |
| 4 | v | $m\ s^{-1}$ | Horizontal wind speed (zero due to rotation of coordinate system) |
| 5 | w | $m\ s^{-1}$ | Vertical wind speed |
| 6 | Ts | °C | Sonic temperature |
| 7 | Tp | °C | [no data] |
| 8 | a | $g\ m^{-3}$ | $H_2O$ content of the air (LI7500) |
| 9 | CO2 | $mmol\ m^{-3}$ | $CO_2$ content of the air |
| 10 | T_ref | °C | Air temperature |
| 11 | a_ref | $g\ m^{-3}$ | Reference $H_2O$ content of the air (HMP45C) |
| 12 | p_ref | hPa | Air pressure |
| 13 | Var[u] | $m^2\ s^{-2}$ | Variance of horizontal wind speed |
| 14 | Var[v] | $m^2\ s^{-2}$ | Variance of horizontal wind speed |
| 15 | Var[w] | $m^2\ s^{-2}$ | Variance of vertical wind speed |
| 16 | Var[Ts] | $°C^{-2}$ | Variance of sonic temperature |
| 17 | Var[Tp] | $°C^{-2}$ | [no data] |
| 18 | Var[a] | $g^2\ m^{-6}$ | Variance of $H_2O$ content of the air |
| 19 | Var[CO2] | $mmol^2\ m^{-6}$ | Variance of $CO_2$ content of the air |
| 20 | Cov[u'v'] | $m^2\ s^{-2}$ | Covariance of wind components u and v |
| 21 | Cov[v'w'] | $m^2\ s^{-2}$ | Covariance of wind components v and w |
| 22 | Cov[u'w'] | $m^2\ s^{-2}$ | Covariance of wind components u and w |
| 23 | Cov[u'Ts'] | $°C\ m\ s^{-1}$ | Covariance of wind component u and sonic temperature |
| 24 | Cov[v'Ts'] | $°C\ m\ s^{-1}$ | Covariance of wind component v and sonic temperature |
| 25 | Cov[w'Ts'] | $°C\ m\ s^{-1}$ | Covariance of wind component w and sonic temperature |
| 26 | Cov[u'Tp'] | $°C\ m\ s^{-1}$ | [no data] |
| 27 | Cov[v'Tp'] | $°C\ m\ s^{-1}$ | [no data] |
| 28 | Cov[w'Tp'] | $°C\ m\ s^{-1}$ | [no data] |
| 29 | Cov[u'a'] | $g\ s^{-1}\ m^{-2}$ | Covariance of wind component u and $H_2O$ content of the air |
| 30 | Cov[v'a'] | $g\ s^{-1}\ m^{-2}$ | Covariance of wind component v and $H_2O$ content of the air |
| 31 | Cov[w'a'] | $g\ s^{-1}\ m^{-2}$ | Covariance of wind component w and $H_2O$ content of the air |





| 32 | Cov[u'CO2'] | mmol m$^{-2}$ s$^{-1}$ | Covariance of wind component u and $CO_2$ content of the air |
|----|-------------|------------|---------------------------------------------------------|
| 33 | Cov[v'CO2'] | mmol m$^{-2}$ s$^{-1}$ | Covariance of wind component v and $CO_2$ content of the air |
| 34 | Cov[w'CO2'] | mmol m$^{-2}$ s$^{-1}$ | Covariance of wind component w and $CO_2$ content of the air |
| 35 | Nvalue | - | Number of samples the aggregated 30-min-value is based on |
| 36 | dir | ° | Wind direction |
| 37 | ustar | m s$^{-1}$ | Friction velocity |
| 38 | HTs | W m$^{-2}$ | Sensible heat flux |
| 39 | HTp | W m$^{-2}$ | [no data] |
| 40 | LvE | W m$^{-2}$ | Latent heat flux |
| 41 | z/L | - | Stability parameter (positive values denote stable boundary layer) based on sonic temperature |
| 42 | z/L-virt | - | Stability parameter (positive values denote stable boundary layer) based on virtual temperature |
| 43 | Flag(ustar) | - | Quality flag for ustar time series. Refer to the flag info in the "general info" sheet |
| 44 | Flag(HTs) | - | Quality flag for sensible heat time series. Refer to the flag info in the "general info" sheet |
| 45 | Flag(HTp) | - | [no data] |
| 46 | Flag(LvE) | - | Quality flag for latent heat time series. Refer to the flag info in the "general info" sheet |
| 47 | Flag(wCO2) | - | Quality flag for NEE time series. Refer to the flag info in the "general info" sheet |
| 48 | T_mid | - | UTC-time middle of interval (YYYY-MM-DD hh:mm) |
| 49 | FCstor | mmol m$^{-2}$ s$^{-1}$ | $CO_2$ storage of the air column below the measurement height |
| 50 | NEE | mmol m$^{-2}$ s$^{-1}$ | Net ecosystem exchange of $CO_2$ |
| 51 | Ftprint_trgt_1 | % | Cumulative source contribution of the target area |
| 52 | Ftprint_trgt_2 | % | Cumulative source contribution of adjacent areas of the same type as the target area |
| 53 | Ftprnt_xmax | m | Distance between EC-tower and the point of the maximum source contribution |
| 54 | r_err_ustar | % | Relative random error of ustar |
| 55 | r_err_HTs | % | Relative random error of HTs |

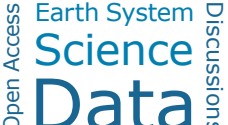

| 56 | r_err_LvE | % | Relative random error of LvE |
|----|-----------|---|------------------------------|
| 57 | r_err_co2 | % | Relative random error of $CO_2$ flux |
| 58 | noise_ustar | % | Relative noise error of ustar |
| 59 | noise_HTs | % | Relative noise error of HTs |
| 60 | noise_LvE | % | Relative noise error of LvE |
| 61 | noise_co2 | % | Relative noise error of $CO_2$ flux |

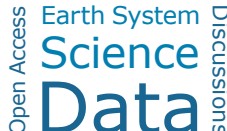

**Table A4: Size ranges of particle size classes (DIN 14688)**

| particle size class | abbreviation | size range (mm) |
|---|---|---|
| coarse material / gravel | Gr | > 2 |
| coarse sand | CSa | 0.63–2 |
| medium sand | MSa | 0.2–0.63 |
| fine sand | FSa | 0.063–0.2 |
| sand | Sa | 0.063–2 |
| coarse silt | CSi | 0.02–0.063 |
| medium silt | MSi | 0.0063–0.02 |
| fine silt | FSi | 0.002–0.0063 |
| silt | Si | 0.002–0.063 |
| clay | Cl | ≤ 0.002 |





**Table A5: Columns of soil datafiles. For particle size classes see Table A4.**

| #Col. | variable | units | datatype | Description |
|---|---|---|---|---|
| 1 | Site | - | character | Site ID |
| 2 | Field | - | character | Field ID (per site) |
| 3 | UTM_northing | m | numeric | UTM Northing (WGS84, 32N) |
| 4 | UTM_easting | m | numeric | UTM Easting (WGS84, 32N) |
| 5 | Depth | cm | character | Sampling depth (layer) |
| 6 | horizon | - | character | Soil horizon (see Schad et al., 2009) |
| 7 | CSa | mass% | numeric | Percentage of coarse sand particles in fine soil |
| 8 | MSa | mass% | numeric | Percentage of medium sand particles in fine soil |
| 9 | FSa | mass% | numeric | Percentage of fine sand particles in fine soil |
| 10 | Sa | mass% | numeric | Percentage of sand particles in fine soil (CSa+MSa+FSa) |
| 11 | CSi | mass% | numeric | Percentage of coarse silt particles in fine soil |
| 12 | MSi | mass% | numeric | Percentage of medium silt particles in fine soil |
| 13 | FSi | mass% | numeric | Percentage of fine silt particles in fine soil |
| 14 | Si | mass% | numeric | Percentage of silt particles in fine soil (GSi+MSi+FSi) |
| 15 | Cl | mass% | numeric | Percentage of clay particles in fine soil |
| 16 | date_part_siz | - | date and time | sampling date for particle size distribution (in the field, YYYY-MM-DD) |
| 17 | Gr | mass% | numeric | Percentage of coarse material / gravel in soil sample |
| 18 | bulk_dens | g cm$^{-3}$ | numeric | Bulk density |
| 19 | date_bulk_dens | - | date and time | sampling date for bulk density (in the field, YYYY-MM-DD) |
| 20 | SOC | mass% | numeric | Soil organic carbon content |
| 21 | tot_C | mass% | numeric | Total carbon content |
| 22 | tot_N | mass% | numeric | Total nitrogen content |
| 23 | date_CN | - | date and time | Sampling date for C- and N-content (in the field, YYYY-MM-DD) |
| 24 | soil_unit | - | character | Assignment to a soil unit of Brogi et al. (2019), only Selhausen |
| 25 | comment | - | character | Comment |





**Table A6: Columns of the soil units datafile. This data exists for the site Selhausen only. For particle size classes see Table A4.**

| #Col. | variable | units | datatype | description |
|---|---|---|---|---|
| 1 | soil_unit | - | character | Assignment to a soil unit of Brogi et al. (2019) |
| 2 | horizon | - | character | Soil horizon (see Schad et al., 2009) |
| 3 | max_depth | cm | numeric | Maximum depth of the soil horizon found in the corresponding soil unit |
| 4 | Sa | mass% | numeric | Percentage of sand particles in fine soil |
| 5 | Si | mass% | numeric | Percentage of silt particles in fine soil |
| 6 | Cl | mass% | numeric | Percentage of clay particles in fine soil |
| 7 | Gr | mass% | numeric | Percentage of coarse material / gravel in soil sample |
| 8 | tot_C | mass% | numeric | Total carbon content |
| 9 | tot_N | mass% | numeric | Total nitrogen content |
| 10 | CV_max_depth | % | numeric | Uncertainty of max_depth (coefficient of variation) |
| 11 | CV_Sa | % | numeric | Uncertainty of Sa (coefficient of variation) |
| 12 | CV_Si | % | numeric | Uncertainty of Si (coefficient of variation) |
| 13 | CV_Cl | % | numeric | Uncertainty of Cl (coefficient of variation) |
| 14 | CV_Gr | % | numeric | Uncertainty of Gr (coefficient of variation) |



**Table A7: Columns of weather datafiles. With the exception of the timestamps, all datatypes are numeric.**

| #Col. | variable | units | Description |
|---|---|---|---|
| 1 | Date & Time begin (UTC) | - | UTC-time beginning of interval (YYYY-MM-DD hh:mm) |
| 2 | Date & Time end (UTC) | - | UTC-time end of interval (YYYY-MM-DD hh:mm) |
| 3 | AirPres | Pa | Air pressure |
| 4 | GlobRad | W m$^{-2}$ | Global Radiation |
| 5 | AirTemp | K | Air temperature |
| 6 | AirHum | % | Relative humidity of the air |
| 7 | Wind | m s$^{-1}$ | Wind speed |
| 8 | Precip | Mm | Precipitation |
| 9 | SurfaceTemp | K | Surface temperature* |
| 10 | InLW | W m$^{-2}$ | Incoming longwave radiation |
| 11 | Cloudiness | 1/8 | Cloud cover |

\* contains no data, included for compatibility purposes