# Peer review of "A comprehensive dataset of vegetation states, fluxes of matter and energy, weather, agricultural management, and soil properties from intensively monitored crop sites in Western Germany"

_Earth System Science Data, 2019_

## Referee Comment (RC1) · Anonymous Referee #1 · 27 Apr 2020

I appreciate that the editor has offered this opportunity to comment. In this manuscript, Reichenau et al. present a comprehensive dataset of vegetation states, fluxes of matter and energy, weather, agricultural management, and soil properties from four intensively monitored crop sites in Western Germany. The authors are to be commended for undertaking an ambitious project and providing types of data useful to the agriculture and biogeochemical modeling communities. In this manuscript, all kinds of information regarding the development of the dataset are demonstrated. I especially appreciate that the authors specified the detailed uncertainty and quality assurance of each dataset,

which is not usually seen in many published datasets. I think this dataset and paper are worth publishing in ESSD. However, there are some major or minor information needs to be clarified before its acceptance and publication.

1. These sites are/were intensively agricultural sites. As an ecosystem modeler, considering the purpose of initializing crop/ecosystem models, the pre-management vegetation and soil conditions (e.g., soil C and N content, potential vegetation type and its states) should be used to initialize models. However, the data regarding vegetation and soil properties provided in this paper are quite contemporary, say time coverage is from 2007 to 2017. The initial conditions of vegetation and soils were missed in these sites. Is it possible to add some of the background information to make the dataset more complete?

2. Under section 5, please specify the beginning and ending date of the 15 time series. I went over the spreadsheet and found the temporal coverages of these time series are not even. Still, it is amazing to see the 30-min interval data!

3. Under section 7, four different gap-filling methods were actually used. However, the reasoning of each method every time it is used at different sites is not explicit enough. Under what circumstances each method was exactly used? And why?

Minor comments:

The current organization of the paragraphs is not good enough. There is no first-line indentation where it is needed. Bullet points also do not have indentations. Some sloppy spelling and wrongly used punctuation and marks. Also, no line number, which makes it hard to specify comments.

Section 2.1 Line 11: within the parenthesis, % was used while degree was used in other places. Be consistent. Line 19: data is a plural term!

Section 2.2 Line 25, degree sign is used here. Be consistent.

Section 2.3 Last line: what is the temporal range for the mean annual temperature of

9.7 degrees in this site?

Section 3 Line 10: data is a plural term! Line 15: remove "and" before "can thus be"

Section 4.1 Line 4: remove "given"

Section 5.1 Line 20: please use subscript for $H_2O$ and $CO_2$.

Section 7.1.1 Line 30: please specify the years for the sites for which the EOF method was applied.

Page 16 Line 6: please correct the " and " signs.
* * *

---

## Referee Comment (RC2) · Anonymous Referee #2 · 27 Apr 2020

Summary and general comments

Reichenau et al. presents a comprehensive dataset related to vegetation states, energy and water fluxes, agricultural management, and soil properties from four agricultural sites in Western Germany. I appreciate the efforts the authors made in detailing all these datasets; however, there are also some confusions and issues that the authors should clarify in further revisions.

(1) Scientific relevance: As the selected sites are from the same area and are all

agricultural-based sites, I would suggest the authors further sharpen their research motivation by stating how these datasets can facilitate understanding agricultural ecosystems and then generalize to the general ecosystem. Right now, it seems we can use these datasets to understand all aspects of ecological settings as the introduction specifies. In addition, as flux datasets from eddy covariance flux towers are also available, the authors need to point out what are the new items we can find in this dataset. What is missing before and what would be new data that are important to further gain insights of ecological systems? In addition, the provision of these data from only this area also raises the question whether these data are representative to investigate issues such as impacts of climate change on agricultural ecosystems? The authors may also want to clarify.

(2) Datasets: there are quite many missing data shown in Excel as NA. I am not sure whether it makes the whole data series. The authors also need to document why these missing data are there and if gap filling algorithms were used (at the least the manuscript provides gap filling algorithms) and/or needs to be used. In addition, data documents from all these sites are not consistent, for example, EC folder in all the sites except huertgenwald. Mangement data from the merken site only have a couple of years data. Why some years of data are missing? Some sites do not have meteo.csv? Overall, the data are not as detailed as those described in the manuscript. It is also hard to understand some of the excel documents since no metadata are provided to explain the variables? For example, what is u, v, w in the fluxes_MEF01_EC_001_2011?

(3) While the authors attempt to detail the datasets, it is necessary for the authors to provide at least several application examples that how these datasets can be used. Particularly, as the authors trying to understand "patterns in soil-vegetation-atmosphere-systems", it would be helpful to see some research results from this dataset. This would also help clarify how these datasets can be used even if there are missing data. I do not expect complicated application examples, but generally for example vegetation states and temperature/precipitation can be used to understand

climate impacts on agricultural ecosystems.

---

## Referee Comment (RC3) · Anonymous Referee #3 · 28 Apr 2020

The authors generated a comprehensive dataset at four agricultural sites for the development and validation of hydro-ecological land-surface models, and as well as the remote sensing products. Thank the authors for the efforts on the data collection. As the authors stated that one of the goals of this dataset is for the validation of the remote sensing products, there are still some places that need the authors' clarification and consideration. I recommend a minor revision before publication. Section 4, P5 L14: The leaf area index was measured by the destructive approach at a very small sampling area, e.g., 40 x 40 cm. I fully understand that this was due to the limitation

of labor, while the spatial representativeness of the measurements can make the direct validation of the remote sensing products (10 m∼ 1km) to be challenging. The inhomogeneity of the vegetation states over the study area may need to be evaluated by the UAV data or very high spatial resolution data if available, especially at the same periods of the measurements. The corresponding high-resolution data can be very important for bridging the gap between the point-scale field measurements and satellite-level remote sensing products, as the two websites below. Even the information about the availability of the high-resolution data with good quality can also be very helpful for the users. http://w3.avignon.inra.fr/valeri/fic_htm/database/main.php http://calvalportal.ceos.org/web/olive/site-description

Besides, the temporal frequency of the LAI measurements collected from 2007 to 2017 may need to be clearly explained in the manuscript for the readers. The publicly available LAI measurements at the seasonal cycle are rare and valuable for the validation of the current remote sensing products in time series.

For the linkage of the vegetation and soil properties with remote sensing data, in addition to the canopy structure parameters such as LAI, not quite sure whether there are also leaf and soil spectrum, leaf chlorophyll and water content, and photos of the vegetation canopy at different growing stages available. If yes, these data would be very useful for the radiation transfer modeling and analysis over the agricultural ecosystem.

---

## Referee Comment (RC4) · Anonymous Referee #4 · 28 Apr 2020

I appreciate the authors' efforts in compiling this dataset. As a modeler, I understand the importance of such a dataset for model parameterization and validation. Data collection and compiling are usually two of the most time-consuming processes. It's good to see this dataset been organized following a consistent protocol. However, the importance of the current dataset is not clear. Such data (e.g. flux, managements) can be found from different sources. For example, Ameriflux (https://ameriflux.lbl.gov/) provides multi-year, quality-controlled data collected from hundreds of eddy-flux sites worldwide, and the management information for the cropland sites can be obtained

from the website and the related publications. The authors did provide a description of the uniqueness of the dataset, but it sounds the data has limited impacts on at local area.

Honestly, I am also not very satisfied with many missing data (NAs) in this dataset, especially in the management information category. The dataset barely provides a three-year continuous management records. I understand that this is labor-intensive work, but the current dataset is relatively short-term coverage. Are the authors planning to continue the measurements and regularly update the dataset?

Another concern is the lack of land use history information. For example, land use change plays a pivotal role in altering soil property. How long have the sites been converted to cropland? Generally, conversion from natural vegetation to cropland triggers rapid decomposition of soil organic matters, which may last for decades. Have the lagged impacts from land use change gone? This background information is essential, which should be provided before the data can be used in modeling.

---

## Short Comment (SC1) · 6 May 2020

This manuscript provides a comprehensive dataset collected at four agricultural sites within the Rur catchment in Western Germany. The dataset includes various variables, e.g., vegetation states, carbon and energy fluxes, meteorological variables, agricultural management, and soil properties. The topic is an important one and the manuscript is well written. Major Comments 1. The authors have provided detailed description of the development and information about the dataset, e.g., the quality flags and the uncertainty. However, a discussion/conclusion section is missing, where the main ad-

vantages/disadvantages or the cautions in using the dataset of the provided dataset should be highlighted to the readers. I believe that highlight would greatly benefit the potential users of the dataset. 2. The manuscript seems incomplete with reference section missing. Some basic methods or theory used in the data development process should be cited, e.g., the barometric formula used in Page 16 Line 30.

Minor Comments

1. Page 2 Line 19. Why use capital letters in "Monitoring, Modelling and Data"?

2. Page 2 Line 21. "TERENO (Terrestrial Environmental Observatories)" the abbreviation should be in the bracket?

3. Page 3 Line 12. What is "WRB"? please explain.

4. Page 3 Line 12. "carbon (NEE), water (LE), and energy (H)", please explain the meaning of the abbreviations first before using them, e.g., LE the latent heat flux, H the sensible heat flux.

5. Page 11 Line 21. TYPO. "CO2" should be "$CO_2$". Please also check throughout the manuscript.

6. Page 15 Line 11. TYPO. "On short timescales"?

7. Page 16 Line 6. TYPO. ""bad", or "suspicious""

8. Page 16 Line 30. Reference should be cited for the barometric formula?

---

## Author Comment (AC1) · 25 Jun 2020

First of all, we are happy, that the comments on the manuscript are generally positive. As some of the reviewers wrote, the compilation and quality control of a dataset like the one described here, really was a lot of work. We want to thank the four anonymous reviewers and Jianxi Huang, who did a short comment, for their statements. Here, we respond to all comments given and explain which changes we applied to the manuscript and dataset. Reviewer comments are black, the authors' response is green and changes to the text are blue.

RC1

I appreciate that the editor has offered this opportunity to comment. In this manuscript, Reichenau et al. present a comprehensive dataset of vegetation states, fluxes of matter and energy, weather, agricultural management, and soil properties from four intensively monitored crop sites in Western Germany. The authors are to be commended for undertaking an ambitious project and providing types of data useful to the agriculture and biogeochemical modeling communities. In this manuscript, all kinds of information regarding the development of the dataset are demonstrated. I especially appreciate that the authors specified the detailed uncertainty and quality assurance of each dataset, which is not usually seen in many published datasets. I think this dataset and paper are worth publishing in ESSD. However, there are some major or minor information needs to be clarified before its acceptance and publication.

Thank you for the positive feedback. It was indeed a lot of work to compile and quality-check everything.

1. These sites are/were intensively agricultural sites. As an ecosystem modeler, considering the purpose of initializing crop/ecosystem models, the pre-management vegetation and soil conditions (e.g., soil C and N content, potential vegetation type and its states) should be used to initialize models. However, the data regarding vegetation and soil properties provided in this paper are quite contemporary, say time coverage is from 2007 to 2017. The initial conditions of vegetation and soils were missed in these sites. Is it possible to add some of the background information to make the dataset more complete?

The whole area of investigation has been inhabited and agriculturally used for more than 2000 years. Because of the very long period of agricultural land-use, it can be assumed, that there are no persisting effects of the conversion to agricultural area. Therefore, the soil C- and N-contents shown represent the current states as influenced by current management with the usual variations due to plant growth (seasonality), fertilization, and tillage. The data given in the paper is all measured data we know about.

In the context of a comment in RC4 (see below), we added information on the duration of agricultural use of the sampled fields.

2. Under section 5, please specify the beginning and ending date of the 15 time series. I went over the spreadsheet and found the temporal coverages of these time series are not even. Still, it is amazing to see the 30-min interval data!

For the readers' convenience, we added the periods of the EC timeseries to Table 5. Furthermore, the datafiles were changed. They now do always start at 00:00 of the first day and end at 23:59 of the last day.

3. Under section 7, four different gap-filling methods were actually used. However, the reasoning of each method every time it is used at different sites is not explicit enough. Under what circumstances each method was exactly used? And why?

In the course of the project, different methods were applied to fill the gaps in the meteorological data. There were two project-wide efforts for meteo gap-filling. The latest used the EOF method. This is the method with the best evaluation. Therefore, this data was used where available. It could not

be applied earlier than 2011 since the number of available stations was too small then. In the earlier years, first the insertion method and later the more sophisticated regression method 2 was applied, which reflects a methodological progress in the project. Gap-filling in Hürtgenwald was not done with the EOF method, since this station does not belong to the centrally maintained stations of the project.

To make this clear to the readers, short explanations were added in section 7.1.1.

Minor comments:

The current organization of the paragraphs is not good enough. There is no first-line indentation where it is needed. Bullet points also do not have indentations. Some sloppy spelling and wrongly used punctuation and marks. Also, no line number, which makes it hard to specify comments.

We are sorry for the inconvenience experienced by reviewer 1. In general, we used the template provided by the Journal. However, we have to admit, that we were not consequent with the formatting of enumerations and bullet lists. This will be corrected during the typesetting process of the revised manuscript (if accepted for publication) as will be the indentation. We have no explanation why the line numbers were missing. They are present in the pdf-document on the journal's website.

We will check the revised manuscript for compliance with the journal's typesetting guidelines.

Section 2.1 Line 11: within the parenthesis, % was used while degree was used in other places. Be consistent.

We thank the reviewer for the thorough review!

We changed the units accordingly.

Line 19: data is a plural term!

Here, we would like to refer to http://dataabinitio.com/?p=497 and point to the third paragraph. The author points out that, due to evolution of language, data nowadays can be used as singular. Since there are about three times more google hits with the singular form, we kept this.

Section 2.2 Line 25, degree sign is used here. Be consistent.

This was unified for comment Section 2.1 line 11 above.

Section 2.3 Last line: what is the temporal range for the mean annual temperature of 9.7 degrees in this site?

The temporal range for this data is unknown. It was taken from Schulz (2004) since this was the only literature with data on Merzenhausen climate. Unfortunately, there is no way to ask Elke Schulz for details on the data, because she retired recently and has no public contact address.

We added the citation to the sentence on climate.

Section 3 Line 10: data is a plural term!

See above.

Line 15: remove "and" before "can thus be"

Sentence was rephrased

Section 4.1 Line 4: remove "given"

done

Section 5.1 Line 20: please use subscript for H2O and CO2.

done

Section 7.1.1 Line 30: please specify the years for the sites for which the EOF method was applied.

Already done in context of major comment 3.

Page 16 Line 6: please correct the " and " signs.

done

RC2

Summary and general comments

Reichenau et al. presents a comprehensive dataset related to vegetation states, energy and water fluxes, agricultural management, and soil properties from four agricultural sites in Western Germany. I appreciate the efforts the authors made in detailing all these datasets; however, there are also some confusions and issues that the authors should clarify in further revisions.

Thank you for the positive feedback.

(1) Scientific relevance: As the selected sites are from the same area and are all agricultural-based sites, I would suggest the authors further sharpen their research motivation by stating how these datasets can facilitate understanding agricultural ecosystems and then generalize to the general ecosystem. Right now, it seems we can use these datasets to understand all aspects of ecological settings as the introduction specifies. In addition, as flux datasets from eddy covariance flux towers are also available, the authors need to point out what are the new items we can find in this dataset. What is missing before and what would be new data that are important to further gain insights of ecological systems? In addition, the provision of these data from only this area also raises the question whether these data are representative to investigate issues such as impacts of climate change on agricultural ecosystems? The authors may also want to clarify.

We respond to this comment and to a comment in RC4 where the reviewer states "the importance of the current dataset is not clear".

The intention of the compilation and publication of this dataset is to provide coherent and comprehensive data on agricultural land. It is intended to provide modelers and remote sensing scientists a set of relevant types of data collected in a 12-year project. Parameterized models or remote sensing algorithms resulting from the usage of the data can then be used for whatever purpose the model or algorithm is applied to. This encompasses everything that requires a good representation of vegetation, be it studies on hydrology, balances of energy or mass, food production, etc. A generalization to non-agricultural ecosystems was not in our scope. Furthermore, the data is not assumed to be representative for agricultural ecosystems in general, since these show a large variability in terms of crops, management, soil, and climate. Therefore, detailed data from agricultural ecosystems differing in the named properties are required for large-scale analyses, e.g. on the continental or national scale. This illustrates the need for this kind of datasets from different places in order to develop realistic information on the regional differences of states and fluxes. By this means, the whole dataset is the reviewer's "new item".

We added a statement that this kind of data is required from different agricultural regions in the introduction.

Since it is usually hard to find fitting data for the same sites, we wanted to provide a convenient overall collection of well-documented quality-controlled data, which is ready to use. Let's take the example of a modeler interested in simulating fluxes in a highly fragmented agricultural area with small fields. In order to accomplish this, flux data is required together with data on vegetation, soil, weather, and management, not only for the field with flux measurements but also for surrounding fields for extrapolation purposes. This kind of data is not available from the Fluxnet or Ameriflux networks where ancillary data is (if at all) only offered for the flux-measurement site itself. In this dataset, such data is available. Especially for the Selhausen site, the detailed soil map is a unique feature. Furthermore, this dataset includes six unpublished flux time-series from the site Merken. These are unpublished measurements from three nearby fields measured simultaneously at two heights, a setup which is rarely available.

We added a statement on the previously unpublished two-height flux data for the Merken site and on the differences in data availability between Fluxnet/Ameriflux and this dataset in the introduction and mentioned the former in the abstract.

(2) Datasets: there are quite many missing data shown in Excel as NA. I am not sure whether it makes the whole data series. The authors also need to document why these missing data are there and if gap filling algorithms were used (at the least the manuscript provides gap filling algorithms) and/or needs to be used.

It is true that there are many data-points marked unavailable. One the one hand, this is due to maintaining a consistent structure of the data-tables: (1) If a certain measured variable is available for one site, the corresponding column in the tables is there for all sites. (2) For flux data, we used preexisting file formats, which define columns for magnitudes that were not measured in our case. Both cases cause columns totally filled with NA. On the other hand, some data had to be removed from the dataset due to quality control procedures.

We added this clarification in a new section 3.1.

Gap-filling was solely used for meteorological data because this is required to run a simulation model. Since flux data is sometimes provided with gaps filled, we added a statement in section 5.1 that no gap-filling was applied. We did not add such statements to the sections on vegetation data, because we do not want to start an unnecessary discussion on whether the indirect calculation of a biomass value should be called gap-filling. We consider this unnecessary, since the methods applied are described in detail in section 4.2. In case of soil data, we abstain from adding a statement on gap filling, since the method used by Brogi et al. (2019) to define soil units include an interpolation process. Because it is far from the topic of this paper, and because the methods are described in detail, we do not want to discuss this here.

Since flux data is sometimes provided with gaps filled, we added a statement in section 5.1 that no gap-filling was applied

In addition, data documents from all these sites are not consistent, for example, EC folder in all the sites except huertgenwald. Mangement data from the merken site only have a couple of years data. Why some years of data are missing? Some sites do not have meteo.csv?

Overall, the data are not as detailed as those described in the manuscript.

In this comment, the reviewer states that the documents for the sites were not consistent. From our point of view, all the information the reviewer is missing is given in the text. Since there were no flux measurements in Hürtgenwald, there is no folder for this data. The same applies to meteorological data for Merken as described starting in line 28 in section 7.1. The availability of management data is given per data-type in Table 3.

We extended the statements on data availability in the sections on flux data and crop management data.

It is also hard to understand some of the excel documents since no metadata are provided to explain the variables? For example, what is u, v, w in the fluxes_MEF01_EC_001_2011?

Regarding the variables, the metadata in question is given in Tables A2 and A3, which are referred to in the text in section 5.4.

(3) While the authors attempt to detail the datasets, it is necessary for the authors to provide at least several application examples that how these datasets can be used. Particularly, as the authors trying to understand "patterns in soil-vegetationatmosphere-systems", it would be helpful to see some research results from this dataset. This would also help clarify how these datasets can be used even if there are missing data. I do not expect complicated application examples, but generally, for example vegetation states and temperature/precipitation can be used to understand climate impacts on agricultural ecosystems.

The paper includes several references to publications where the data has been used. These papers illustrate several applications of the data, e.g.

LAI: Brogi et al. (2020), Reichenau et al. (2017), Ney and Graf (2018), Ahrends et al. (2014)

biomass and phenology: Schmidt et al. (2012), Korres et al. (2013)

management: Schmidt et al. (2012), Korres et al. (2013)

weather: Schmidt et al. (2012), Korres et al. (2013), Sakai (2016)

fluxes: Klosterhalfen et al. (2017), Schmidt et al. (2012), Ney and Graf (2018), Ahrends et al. (2014), Wienecke et al. (2018), Eder et al (2015)

soil: Borneman et al. (2011), Korres et al. (2013), Jakobi et al. (2020), Meyer et al. (2017)

For complete references please see the manuscript.

References to papers where the data was used were added to the end of each data-source section.

RC3

The authors generated a comprehensive dataset at four agricultural sites for the development and validation of hydro-ecological land-surface models, and as well as the remote sensing products. Thank the authors for the efforts on the data collection. As the authors stated that one of the goals of this dataset is for the validation of the remote sensing products, there are still some places that need the authors' clarification and consideration. I recommend a minor revision before publication.

Thank you for the comment from a remote sensing point of view.

Section 4, P5 L14: The leaf area index was measured by the destructive approach at a very small sampling area, e.g., 40 x 40 cm. I fully understand that this was due to the limitation of labor, while the spatial representativeness of the measurements can make the direct validation of the remote sensing products (10 m_ 1km) to be challenging. The inhomogeneity of the vegetation states over the study area may need to be evaluated by the UAV data or very high spatial resolution data if available, especially at the same periods of the measurements. The corresponding high-resolution data can be very important for bridging the gap between the point-scale field measurements and satellite-level remote sensing products, as the two websites below. Even the information about the availability of the high-resolution data with good quality can also be very helpful for the users. http://w3.avignon.inra.fr/valeri/fic_htm/database/main.php , http://calvalportal.ceos.org/web/olive/site-description

As the reviewer states correctly, the limited area sampled is due to the limited work capacity together with the aim to sample multiple points per field or multiple fields. Furthermore, farmers would not permit to harvest the area of several 10 m pixels (100 m² each). However, especially in the data from the Selhausen site, several sampling points fall within the area of one remote sensing pixel thus enabling an estimate of the variability within the pixel.

In the context of remote sensing, the LAI required for calibration/evaluation is the average LAI of a pixel's area. Based on the above this usually cannot be measured destructively. Unfortunately, no data was collected via UAV. However, a way to connect field measurements to lower-resolution remote sensing products may be made via high resolution remote sensing. E.g. Brogi et al. (2020) calibrated the algorithm of Ali et al. (2015) based on NDVI derived from 5 m resolution RapidEye level 3A data and LAI data from the dataset presented in this manuscript. Reichenau et al. (2016) showed that realistic statistical distribution of LAI over a larger area could be derived without calibration. However, in that case ground truth is required to prove this. The resulting 5 m resolution LAI data can then be used to bridge the gap to lower-resolution datasets.

The example was added in the section on vegetation measurements.

Besides, the temporal frequency of the LAI measurements collected from 2007 to 2017 may need to be clearly explained in the manuscript for the readers. The publicly available LAI measurements at the seasonal cycle are rare and valuable for the validation of the current remote sensing products in time series.

The LAI data has been collected with frequencies from one to three weeks during the growing season. In the years from 2015 to 2017, this was done on overflight days of Sentinel 1 and Radarsat 2. There are also entries for dates without vegetation. These were included to have a zero-point for the calibration of remote sensing algorithms.

This information has been added to the LAI section. A statement on data frequency and measurements on satellite overflight days was added to the abstract.

For the linkage of the vegetation and soil properties with remote sensing data, in addition to the canopy structure parameters such as LAI, not quite sure whether there are also leaf and soil spectrum, leaf chlorophyll and water content, and photos of the vegetation canopy at different

growing stages available. If yes, these data would be very useful for the radiation transfer modeling and analysis over the agricultural ecosystem.

Unfortunately, leaf and soil spectra were not obtained. The photos of the canopy taken during the harvest of plants on the fields were taken for documentation purpose only. There was no standardized protocol for this. Therefore, it remains unclear whether they can be applied in the context of remote sensing applications. The vegetation water content, however, can be calculated from the difference of fresh and dry weight of the biomass.

RC4

I appreciate the authors' efforts in compiling this dataset. As a modeler, I understand the importance of such a dataset for model parameterization and validation. Data collection and compiling are usually two of the most time-consuming processes. It's good to see this dataset been organized following a consistent protocol.

Thank you for this positive feedback.

However, the importance of the current dataset is not clear. Such data (e.g. flux, managements) can be found from different sources. For example, Ameriflux (https://ameriflux.lbl.gov/) provides multi-year, quality-controlled data collected from hundreds of eddy-flux sites worldwide, and the management information for the cropland sites can be obtained from the website and the related publications. The authors did provide a description of the uniqueness of the dataset, but it sounds the data has limited impacts on at local area.

Please see our response to comment (1) of reviewer 2 (RC2), where we included the response on this comment.

Honestly, I am also not very satisfied with many missing data (NAs) in this dataset, especially in the management information category. The dataset barely provides a three-year continuous management records. I understand that this is labor-intensive work, but the current dataset is relatively short-term coverage.

Regarding the missing management data, this is not due to the time needed to collect it. It is simply due to farmers not reporting their data. Since that was, of course, a voluntary contribution of the farmers, there is no way to get that data. Please also refer to our comments on missing data in the response to reviewer 2 (RC2).

We added section 3.1 as a combined response to this comment and RC2.

Are the authors planning to continue the measurements and regularly update the dataset?

Unfortunately, the project has ended and there are no follow-up projects funding the continuation of the sampling. Solely field F01 in Selhausen is undergoing ongoing sampling since it has become an ICOS site.

Another concern is the lack of land use history information. For example, land use change plays a pivotal role in altering soil property. How long have the sites been converted to cropland? Generally, conversion from natural vegetation to cropland triggers rapid decomposition of soil organic matters, which may last for decades. Have the lagged impacts from land use change gone? This background information is essential, which should be provided before the data can be used in modeling.

The area of the Rur catchment is ancient agricultural land. There has been agriculture in the region in the time of the Romans or even earlier. Close to the field in Merzenhausen, a grave from the Bronze Age has been found. Therefore, it is justified to assume that effects of conversion to arable land do not persist.

We added general information on this in the section 2. More detailed statements were added in the subsections of the respective sites.

SC1

This manuscript provides a comprehensive dataset collected at four agricultural sites within the Rur catchment in Western Germany. The dataset includes various variables, e.g., vegetation states, carbon and energy fluxes, meteorological variables, agricultural management, and soil properties. The topic is an important one and the manuscript is well written.

We thank Jianxi Huang for these extra comments.

Major Comments

1. The authors have provided detailed description of the development and information about the dataset, e.g., the quality flags and the uncertainty. However, a discussion/conclusion section is missing, where the main advantages/disadvantages or the cautions in using the dataset of the provided dataset should be highlighted to the readers. I believe that highlight would greatly benefit the potential users of the dataset.

When writing the manuscript, we decided not to add a discussion section. We assume that researchers who use a dataset like the on described in this manuscript are familiar with the requirements and caveats in scientific processes like model validation or the calibration of models and remote sensing algorithms. Since the manuscript is already quite long, we think that this is not the right place for a discussion of basic scientific processes.

2. The manuscript seems incomplete with reference section missing.

In the pdf document on the ESSD website, there is a references section beginning on page 21.

Some basic methods or theory used in the data development process should be cited, e.g., the barometric formula used in Page 16 Line 30.

In our opinion, we cited all resources underlying the process of data collection, evaluation, analysis, and quality control. The barometric formula is derived from basic physical laws. It has been widely used for a long time and thus has no specific reference.

Minor Comments

1. Page 2 Line 19. Why use capital letters in "Monitoring, Modelling and Data"?

The capital letter originated from the project's name. Nevertheless, in this place they should be lowercase.

Corrected

2. Page 2 Line 21. "TERENO (Terrestrial Environmental Observatories)" the abbreviation should be in the bracket?

We think it is a question of style either to give the explanation or the acronym in brackets.

3. Page 3 Line 12. What is "WRB"? please explain.

Done

4. Page 3 Line 12. "carbon (NEE), water (LE), and energy (H)", please explain the meaning of the abbreviations first before using them, e.g., LE the latent heat flux, H the sensible heat flux.

This comment refers to page 10 line 17. We used the common symbols for the fluxes in capital letters. However, the symbols can also be interpreted as abbreviations.

Additional explanations were added.

5. Page 11 Line 21. TYPO. "CO2" should be "$CO_2$". Please also check throughout the manuscript.

Corrected

6. Page 15 Line 11. TYPO. "On short timescales"?

Removed

7. Page 16 Line 6. TYPO. ""bad", or "suspicious""

Already corrected based on a comment by reviewer 1

Done

8. Page 16 Line 30. Reference should be cited for the barometric formula?

Please see our response to the major comment 1 above.

---

## Author Response (AR2)

Dear editors,

we are pleased to see our paper accepted for publication in ESSD. It was a lot of work to gather all the data, and to do the quality control and documentation. It is good to see, that relevance of such data is recognized by the community and we hope that the data will be useful and used.

Regarding the author's names, I want to keep my middle name as a middle initial, since this is the way it is set in all my publications and the name has never been published anywhere before.

Some figures were recreated as pdf with R. Due to different algorithms in the R functions for pdf and png, some numbers on the axes have changed. However, the data displayed has remained exactly the same. It may be necessary to change font size in some figures, depending on the size of the figures in the final publication. Please contact me if updated versions are required.

When preparing the file for final publication, the following errors or inconsistencies were found and corrected:

1. P1L24, P16L22, P16L25, P17L1, P19L21, caption Table 12: corrected time series (space)
2. P5L11: full stop added
3. P7L27, P14L24, P16L28: changed comma to full stop
4. P7L25, P14L24: added "and" to the last item of the list of references
5. P17f: added missing full stops in the enumeration
6. P4L3: corrected the sentence
7. P4L16: corrected "form" to "from"
8. P4L23: added "it"
9. P4L16: changed "two" to "three"
10. P5L18: sections  (plural)
11. P5L24: split sentence
12. P7L19: removed "size"
13. P7L22: corrected the sentence
14. P7L23: distributions (plural)
15. P8L22: obviously
16. P9L9: Fig. (abbreviation)
17. P9L28: additional reference to figure
18. P11L26: towers (plural)
19. P13L4: added missing underscore
20. P14L29: added spaces before %
21. P15L14: added space after full stop
22. P15L15: removed duplicate sentence
23. P16L6: corrected year specification (2010)
24. P17L5: capitalized "TR"
25. P17L6: changes "set" to "site"
26. P18L18: removed parenthesis (duplication)
27. P18L21: lowercased "station"
28. P18L28, P18L30: added parenthesis in figure panel references
29. P22L1: corrected section number (6.4)
30. P22L5: completed appendix information (added table a4)
31. Figure 5: Corrected the caption
32. Table 3: added missing comma in the caption
33. Table 4: corrected font size
34. Table 5: asterisk for the footnote of column "height" was moved from units to column name

35. Table A1: corrected font size of footnote
36. Table 11: lowercased "relative humidity", "error", and "vector"

Kind regards,

Tim Reichenau